# Ketamine evoked disruption of entorhinal and hippocampal spatial maps

Francis Kei Masuda [ORCID][1], Emily A. Aery Jones[1,2], Yanjun Sun[1,2] & Lisa M. Giocomo [ORCID][1] ✉

Ketamine, a rapid-acting anesthetic and acute antidepressant, carries undesirable spatial cognition side effects including out-of-body experiences and spatial memory impairments. The neural substrates that underlie these alterations in spatial cognition however, remain incompletely understood. Here, we used electrophysiology and calcium imaging to examine ketamine's impacts on the medial entorhinal cortex and hippocampus, which contain neurons that encode an animal's spatial position, as mice navigated virtual reality and real world environments. Ketamine acutely increased firing rates, degraded cell-pair temporal firing-rate relationships, and altered oscillations, leading to longer-term remapping of spatial representations. In the reciprocally connected hippocampus, the activity of neurons that encode the position of the animal was suppressed after ketamine administration. Together, these findings demonstrate ketamine-induced dysfunction of the MEC-hippocampal circuit at the single cell, local-circuit population, and network levels, connecting previously demonstrated physiological effects of ketamine on spatial cognition to alterations in the spatial navigation circuit.

Ketamine has been used in clinical medicine as a rapid-acting dissociative anesthetic for decades[1,2]. Recently, ketamine has received increased clinical attention due to its approval to rapidly treat depression at sub-anesthetic doses[3–5]. However, despite common clinical uses as an anesthetic and antidepressant, ketamine can induce undesirable side effects associated with spatial cognition, such as dissociation, psychotomimetic effects and spatial memory and navigation impairments[6–11]. Intriguingly, unlike other classes of memory impairing drugs, sub-anesthetic administration of ketamine can also cause people's internal sense of space to change; patients report feeling disconnected from their own body or as if they are watching the situation as an observer[2,6,12,13]. These effects of ketamine on spatial cognition have also contributed to the abuse of ketamine as a party-drug and as a memory-impairing drug to facilitate assault[6–8,12,14,15].

While the neural substrates associated with ketamine's ability to induce dissociation and hallucination-like perception in mice have been recently investigated[16–18], ketamine's actions on the neural circuits that support spatial navigation and memory are not fully understood. Here, we focus on the impact of ketamine on neural coding in the medial entorhinal cortex (MEC)-hippocampal circuit, which contains neural substrates for generating an internal spatial map of the external environment[19–22]. Previous work in this circuit has shown that ketamine significantly disrupts activity at the level of both single neurons and the larger network. At the level of single neurons, previous in vitro work has demonstrated that ketamine changes the firing rates of MEC and CA1 hippocampal neurons[23–25]. At the network level, low doses of ketamine decrease the coherence and synchrony of local field potential measured theta oscillations across the long axis of the hippocampus and increase the power of gamma oscillations[26–28]. However, the potential correspondence between ketamine's influence on single neuron firing rates, network level oscillatory activity and the spatial patterns of functionally defined neurons, such as place or grid cells, in the MEC or hippocampus remain unknown.

Here, we used Neuropixels silicon probes in MEC and 1-photon miniscope imaging in hippocampus to consider ketamine's effects across the levels of single neuron firing rates and spatial firing patterns,

[1]Department of Neurobiology, Stanford University School of Medicine, Stanford, CA 94305, USA. [2]These authors contributed equally: Emily A. Aery Jones, Yanjun Sun. ✉e-mail: giocomo@stanford.edu

neural populations and individual animals. We found that a sub-anesthetic dose of ketamine disrupted stable spatial coding acutely and induced remapping over longer timescales in MEC. This disruption in MEC spatial coding resulted in a loss in the ability of MEC neural activity to predict an animal's spatial position, possibly due to an acute disruption in MEC cell-pair temporal firing rate relationships—a prerequisite for many network-level computational models capable of generating MEC spatial firing patterns[29–33]. Complementary to the effects of ketamine on MEC, ketamine acutely suppressed the activity of hippocampal place cells. Together, our findings reveal how ketamine disrupts and restructures spatial coding in MEC and the hippocampus, raising the possibility that these circuits may contribute to the effects of ketamine on spatial cognition[26].

## Results

### Virtual reality behavior and electrophysiological recordings

To examine the effects of sub-anesthetic ketamine on neural activity in the medial entorhinal cortex (MEC), we recorded neural activity in head-fixed mice as they navigated a virtual reality (VR) linear track (mouse $n = 8$) (Fig. 1a). For all conditions, the VR linear track consisted of five landmarks (i.e. towers) that repeated every 400 cm. To encourage running, mice received a water reward at the end of the track before seamlessly teleporting back to the start of the track (Fig. 1b). For

each recording session, the first 50 trials served as the baseline condition, in which no manipulation was performed (baseline epoch). After trial 50, the VR was paused for ~10 s while the mouse received a blank intraperitoneal (IP) injection (control insertion) before proceeding to run a further 50 trials (control epoch). After trial 100, the VR was briefly paused and the mouse received a 25 mg/kg IP injection of ketamine before proceeding to run 190 trials (ketamine epoch) (Fig. 1c). We selected 25 mg/kg IP for our experiments, as it is the lowest dose at which sensory dissociation has been reported in mice[16,18].

To record neural activity, we acutely inserted Neuropixels silicon probes[34] into the MEC, with each recording session associated with a unique probe insertion—up to six recording sessions per mouse, three recording sessions per hemisphere (Fig. 1d). Using this approach, we were able to record from thousands of cells (i.e. units) across a large portion of the MEC dorsal-to-ventral axis in individual wildtype mice ($n = 3233$ cells; 30 sessions; 8 mice; Supplementary Fig. 1). During the baseline and control epochs, many of the recorded cells demonstrated "spatially stable" firing patterns, in which cells consistently fired at specific VR track positions (Fig. 1e–f, "Methods," and Supplementary Fig. 2).

We first examined the impact of ketamine on running behavior during navigation in the head-fixed VR, as ketamine administration can

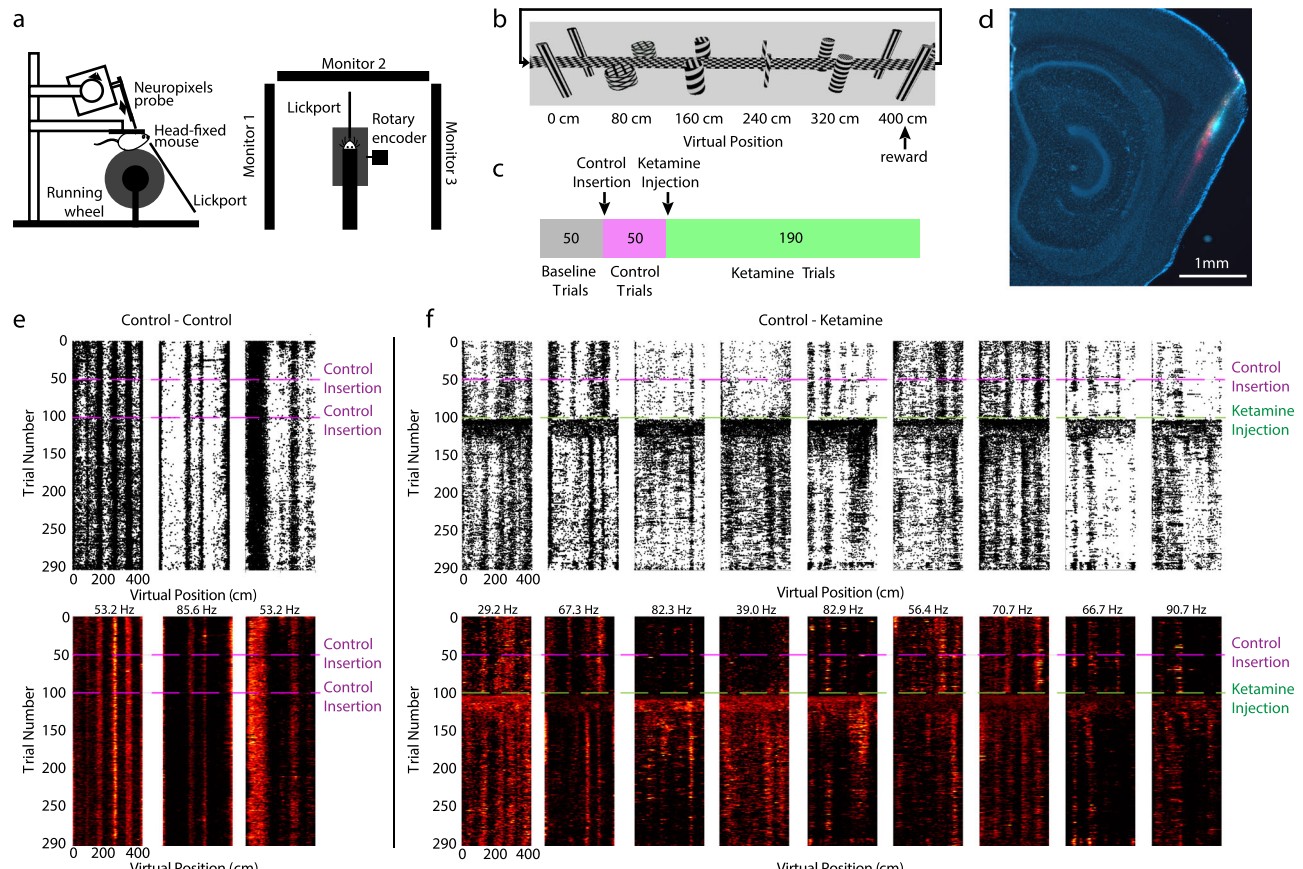

**Fig. 1 | Ketamine alters the coding features of MEC neurons in navigating mice. a** Schematic of Neuropixels probe recording and VR setup. Left: side view. Right: top view. Animals received visual cues from three monitors and received rewards from a front facing lick port. Left panel was published in Low et al.[47], Copyright Elsevier. **b** Schematic of the 400 cm linear VR hallway. Five virtual towers spaced 80 cm apart served as landmarks. One traversal through the hallway is equivalent to one trial. The mouse received a water reward for completing the trial before seamlessly teleporting back to the beginning of the hallway. **c** Animals ran 50 baseline trials (gray), 50 trials following a control needle insertion (pink), and 190 trials following a 25 mg/kg ketamine injection (green). **d** Example sagittal slice of a

mouse brain showing 3 dye color (DiI, DiD, DiO) tracks of Neuropixels probe recording insertions in MEC. Up to 6 recording sessions were conducted on each mouse (3 insertions per brain hemisphere). **e** Spatial raster plots (top row) and spatial firing rate maps (bottom) of example cells. Raster plots indicate individual spikes (black dots). Bottom panels are color coded for minimum (black) and maximum (red) firing rate values. Each cells' maximum firing rate is labeled on the top of the cell's spatial firing rate maps. The panel shows data from the control session in which an empty needle was inserted intraperitoneally after trial 50 and again after trial 100. **f** As in (**e**), for data from the experimental condition of a control injection after trial 50 and a 25 mg/kg ketamine injection after trial 100.

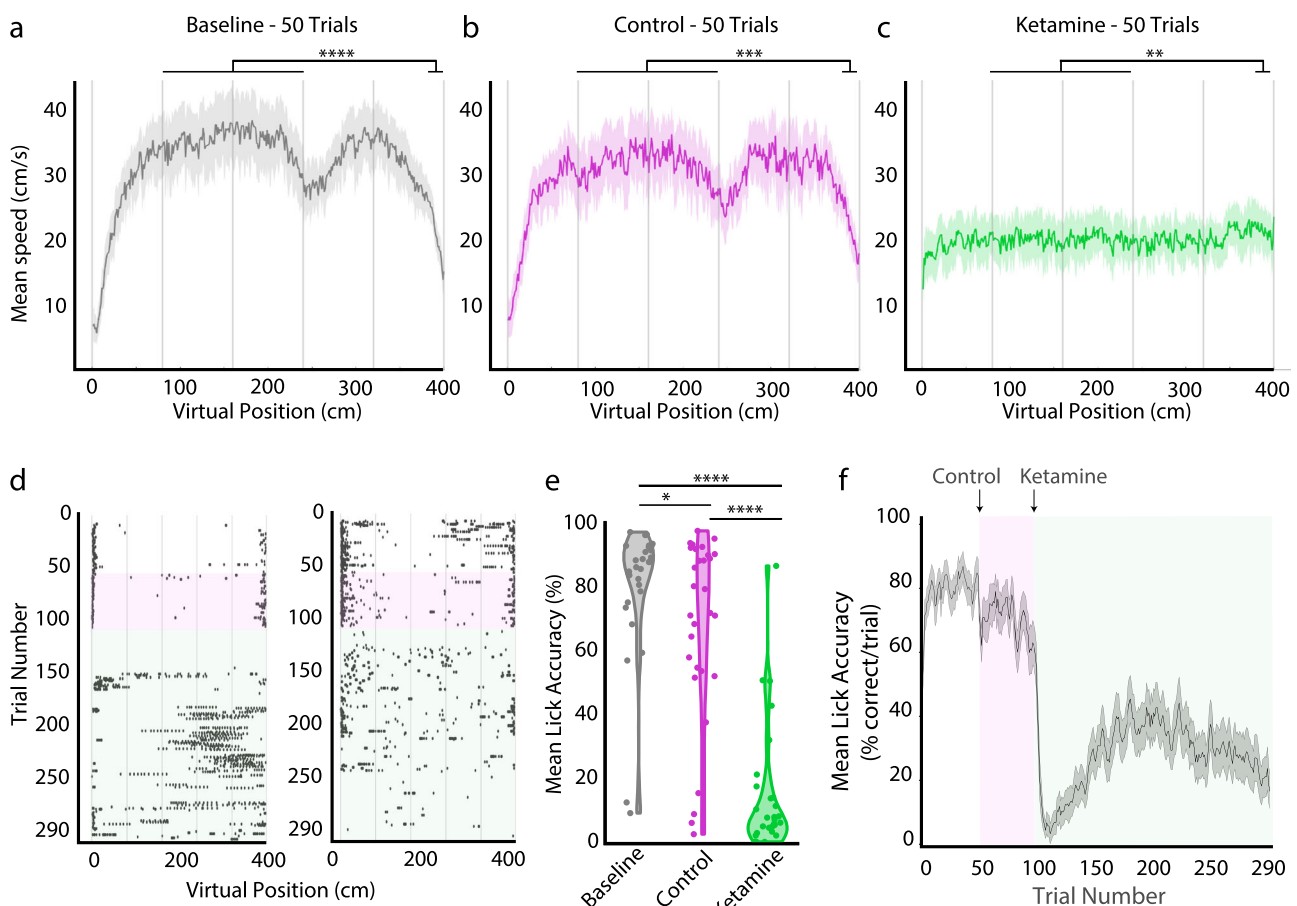

**Fig. 2 | Ketamine disrupts VR task behavior. a** Running speed at each position bin during the baseline epoch (trials 1–50). Gray columns indicate the location of VR landmark towers (spaced by 80 cm). In the reward zone (positions 390–400 cm), mice slowed their running speed compared to the main hallway (100–200 cm) demonstrating familiarity with the task (difference in running speed ± SEM, $16.7 \pm 2.5$ cm/s; $t(29) = 6.77$, $p = 1.96 \times 10^{-7}$). Note the tower at 240 cm resembles the reward tower at 400 cm and mice often slowed down briefly around 240 cm before increasing their speed until they reached the reward tower. **b** Running speed during the control epoch (trials 51–100). In reward zone, mice slowed, demonstrating familiarity with the task ($10.5 \pm 2.4$ cm/s; $t(29) = 4.38$, $p = 0.0001$). **c** Running speed during the first 50 trials of the ketamine epoch (trials 101–150). Mice slightly increased speed in the reward zone ($-2.5 \pm 2.4$ cm/s; $t(29) = -2.84$, $p = 0.0081$). For (**a**–**c**), solid lines represent mean running speed and shaded regions represent SEM.

All tests are two-sided paired $t$ tests. **d** Raster plots of spatial positions where the animal licked in two example recording sessions. Control epoch highlighted in magenta (trials 51–100). Ketamine epoch highlighted in green (trials 101–290). **e** Lick accuracy averaged over sessions in the baseline, control, and ketamine epochs. Lick accuracy was lower in the ketamine epoch than in the baseline ($Z = 4.76$, $p = 5.76 \times 10^{-6}$) and control epochs ($Z = 4.66$, $p = 9.55 \times 10^{-6}$), and lick accuracy in the control epoch was lower than in the baseline epoch ($Z = 2.4$, $p = 0.0489$). Two-sided Wilcoxon matched pairs signed rank tests with Sidak correction for multiple comparisons. **f** Lick accuracy per trial across sessions. Control epoch highlighted in magenta (trials 51–100). Ketamine epoch highlighted in green (trials 101–290). Solid lines represent smoothed lick accuracy and shaded regions represent SEM. $N = 30$ sessions, 8 mice. Significant comparisons highlighted $*p < 0.05$, $***p < 0.001$, $****p < 0.0001$. Source data are provided as a Source data file.

evoke altered behavior during open field exploration (Supplementary Fig. 3). Demonstrating familiarity with and engagement in the task, mice significantly slowed their running speed near the end of the track in anticipation of the reward in both baseline and control conditions (Fig. 2a, b). Following the ketamine injection however, mice no longer significantly slowed down in anticipation of the reward tower and their mean running speed was slower (Fig. 2c). Licking behavior was also altered after the ketamine injection. In the baseline epoch, mice licked near the reward tower with over 80% accuracy (Fig. 2d–f). Licking in the baseline epoch was concentrated around the reward tower, and this behavior remained largely unchanged in the control epoch, with only a small decrease in lick accuracy between the baseline and control epochs (mean accuracy percentage: baseline epoch = 79% ± 1%, control epoch = 69% ±1%; Fig. 2d–f). In contrast, lick accuracy significantly decreased after ketamine administration (mean accuracy percentage: ketamine epoch = 14% ±1%; Fig. 2e, f). Lick accuracy never returned to baseline across the rest of the trials (Fig. 2f and Supplementary Fig. 5a–c). Taken together, these results indicate that while the mice

continued to run in the VR environment after both the control and the ketamine administration, ketamine altered running and licking behavior, suggesting mice were less able to identify the reward location.

We then examined local field potential activity in MEC. Prior work in the hippocampus and cortex has broadly shown that ketamine decreases power in the theta band, while increasing power in the gamma band[27,28,35–37]. Consistent with these previous studies, we observed that ketamine increased power in the theta frequency band but decreased power in the gamma frequency band in MEC (Supplementary Fig. 4). Since theta frequency oscillations have been shown to influence spatial firing patterns in MEC[38–41], we next examined how ketamine changed firing rates and spatial patterns in MEC neurons.

**Ketamine affected firing rates heterogeneously across MEC cell types and impaired spatial coding**

To quantify the effect of ketamine administration of the MEC neural population, we examined the firing rate of all MEC neurons across baseline, control and ketamine epochs. Between baseline and control

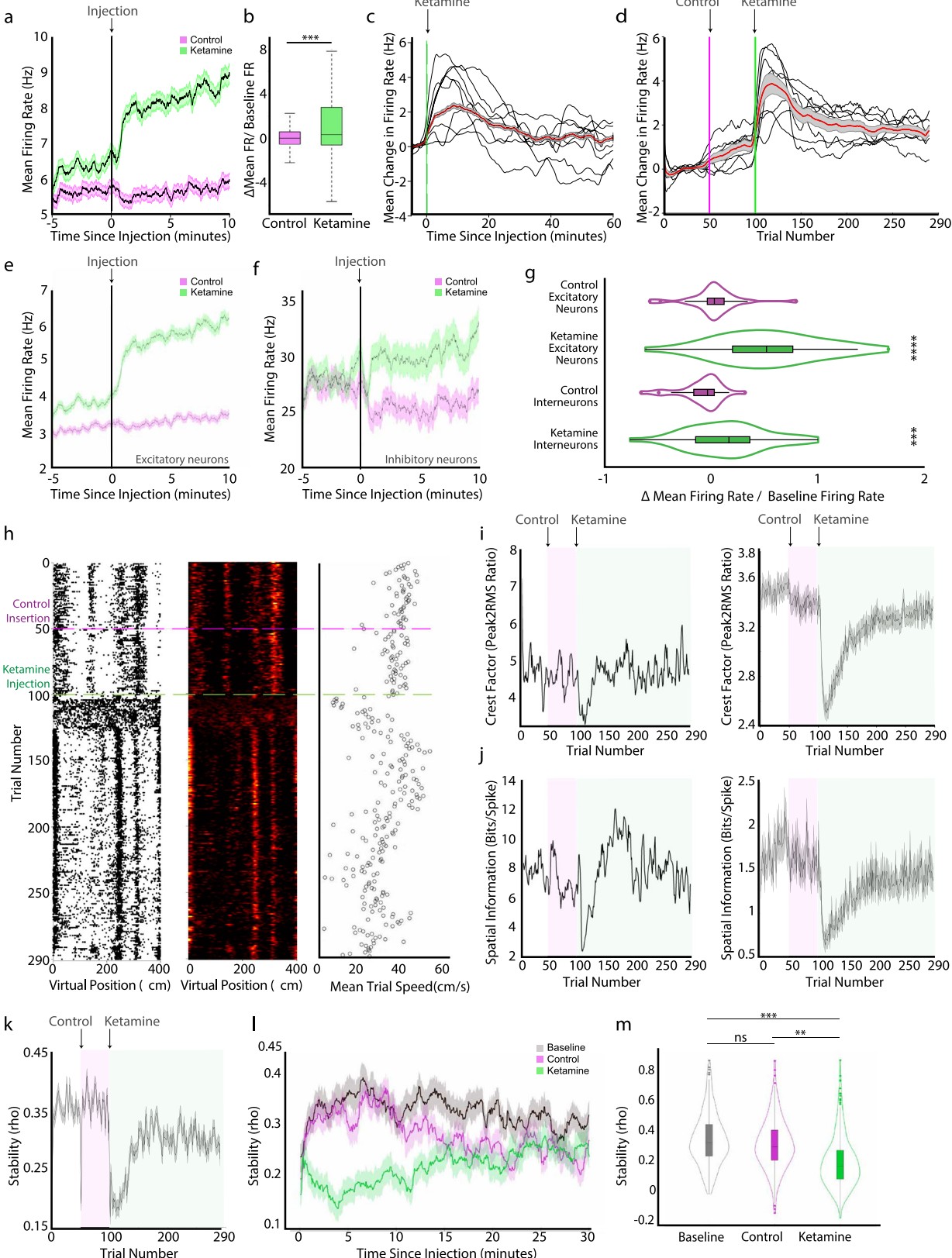

epochs, the mean firing rate across the MEC population only slightly increased (mean change: 0.76 Hz, Wilcoxon signed rank test, $Z = 3.45$, $p = 0.0006$, $n = 30$ sessions), indicating that injection alone did not cause large changes in the firing rates of MEC neurons (Fig. 3a, b). However, during the ketamine epoch, we observed a significant increase in the mean firing rate across the MEC population (mean

change: 3.04 Hz, Wilcoxon matched pairs signed rank test, $Z = 4.29$, $p = 1.8 \times 10^{-5}$, $n = 30$ sessions; Fig. 3a–d). This increase in firing rate in the ketamine epoch was consistent across animals (Fig. 3c, d). The mean peak of the ketamine-induced increase in the firing rate across animals occurred $17.08 \pm 6.26$ min (mean ± SEM) after the injection of ketamine (Fig. 3c). The mean time of firing rate increase onset was

**Fig. 3 | Ketamine acutely affects firing rates and disrupts single cell spatial scores. a** Firing rate averaged over neurons before (5 min) and after (10 min) control (magenta) or ketamine (green) injection. Solid line indicates mean and shaded regions show SEM ($n = 3233$ cells, mouse $n = 8$). Gray line at 0 indicates injection time. **b** Boxplot of mean change in firing rate for the 5 min before versus after the control (magenta) or ketamine (green) injection divided by the baseline firing rate. Central mark indicates the median, bottom and top edges of the box indicate the 25th and 75th percentiles, respectively, and whiskers extend to the most extreme data points not considered outliers. The mean firing rate of neurons changed significantly in the 5 min after ketamine injection compared to the 5 min after control injection (two-sided Wilcoxon matched pairs signed rank test, $Z = -16.45$, $p < 10^{-10}$, $n = 3233$ neurons). **c** Change in the firing rate 60 min after ketamine injection from the mean firing rate 5 min prior to the injection. **d** Change in firing rate across 290 trials from the mean firing rate in the baseline epoch (trial 1–50). Note that due to variable changes in the running speed of individual mice, effects observed over trials (**d**) versus time elapsed (**c**) do not perfectly align. For (**c**, **d**), thin black lines are individual mice averaged across sessions ($n = 8$ mice); thick red line is mean firing rate across all neurons; shaded region shows SEM ($n = 3233$ cells). **e** Same as (**a**), for excitatory neurons ($n = 2894$ cells, mouse $n = 8$). **f** Same as (**a**), for inhibitory neurons ($n = 339$ cells, mouse $n = 8$). **g** Violin plot of the change in the mean firing rate between 5 min before and 5 min after an injection divided by the baseline firing rate of excitatory and inhibitory neurons. Excitatory and inhibitory neurons significantly increase their firing rate (two-sided Wilcoxon

matched pairs signed rank test, excitatory: $Z = 17.57$, $p < 10^{-10}$, $n = 2894$ cells; inhibitory: $Z = 3.6$, $p = 0.0003$, $n = 339$ cells). All violins have the same area, but the width represents the kernel probability density of the data at different values. Central mark of the boxplot indicates the median, bottom and top edges of the box indicate the 25th and 75th percentiles, respectively, and whiskers extend to the most extreme data points not considered outliers. **h** Single example of an MEC spatial cell. Left: Raster plot. Middle: Spatial firing rate map (color coded as in Fig. 1e). Right: Mean running speed per trial. **i** Left: Crest factor for the example cell shown in (**h**). Right: The average crest factor for all cells. **j** Left: Spatial information score for the example cell shown in (**h**). Right: The average spatial information score for all cells. **k** Average spatial stability values. For i-k, solid lines represent mean and shaded regions represent SEM. Control epoch is highlighted in magenta (trials 51–100). Ketamine epoch is highlighted in green (trials 101–300). $N = 300$ trials, 3233 cells, 8 mice. **l** Mean stability of neurons 30 min after baseline (black), control (magenta) and ketamine (green) injection. Line is mean and shaded region is SEM. **m** Comparison of the mean stability of neurons in the baseline (black), control (magenta), and ketamine (green) epochs. Data plotted as violin plots, as in (**g**). Spatial stability slightly differed between the baseline and the control epochs ($Z = 2.93$, $p = 0.003$). The spatial stability of neurons in the ketamine epoch ($n = 50$ trials post ketamine injection) was smaller than in the baseline epoch ($Z = -18.12$, $p < 10^{-20}$) and control epoch ($Z = -17.69$, $p < 10^{-20}$). Two-sided Wilcoxon matched pairs signed rank test, $n = 3233$ cells. Significant comparisons highlighted $**p < 0.01$, $***p < 0.001$, $****p < 0.0001$. Source data are provided as a Source data file.

$3.54 \pm 0.20$ min (mean ± SEM). This coincides with the previously published pharmacodynamics of ketamine (time to onset of immobilization = $2.3 \pm 0.5$ min, serum half-life of ketamine = 13 min)[42] and previously observed effects of ketamine on firing rates in cortical regions[16,43].

We next considered ketamine's effect on the firing rates of different MEC putative-cell types. While the mean rate of excitatory neurons did not change between baseline and the control epochs, their mean firing rates increased after the administration of ketamine ($n = 2894$ excitatory neurons, 8 mice; baseline firing rate <15 Hz; Fig. 3e, g). In contrast, the mean firing rate of interneurons ($n = 339$ interneurons, 8 mice; baseline firing rate >15 Hz) did not appear to change between baseline and control epochs, nor between control and ketamine epochs (Fig. 3f, g). Comparisons between excitatory and inhibitory neurons confirmed these observations, revealing that firing rates of putative excitatory cells significantly increased following ketamine administration, while putative inhibitory cells did not (Fig. 3g).

We then asked whether this increase in excitatory neuron activity was due to disinhibition from local inhibitory neurons or increased intrinsic excitability of excitatory neurons. We identified putative monosynaptically connected pairs of neurons based on their cross-correlograms[44,45], then classified them as excitatory or inhibitory (Supplementary Fig. 6a, b). We calculated the extent to which a single spike from the upstream cell in each pair (cell A) changed the probability of spiking from the downstream cell (cell B) in the 5 ms following the spike, compared to the 5 ms prior to the spike. This transmission probability increased in excitatory neurons and decreased in inhibitory neurons following ketamine injection, thus both excitatory and inhibitory cells were more effective at altering the activity of their downstream partners (Supplementary Fig. 6c–e). This suggests that the increase in excitatory neuron firing was not due to disinhibition, but at least in part due to increased effectiveness of excitatory synaptic transmission.

Given that a large portion of MEC neurons encode the spatial position of an animal, we next considered how ketamine impacted spatial coding across the MEC neural population. In individual MEC neurons, we noted that spatial selectivity often degraded after ketamine administration, switching from a firing pattern in which firing activity occurred in one or more discrete spatial positions on the VR track to a firing pattern in which firing activity was distributed across

the VR track (Figs. 1f and 3h–j). This was consistently observed in spatial cells across many sessions and across different animals (Supplementary Fig. 2). To quantify this, we binned firing rates into 2 cm spatial bins and classified spatial cells as those with a mean spatial stability score >0.2 in the baseline condition ($n = 496$ cells; "Methods"). For spatial cells, there was no change in the mean spatial stability between baseline and control epochs (Fig. 3k–m). This observation held when comparing spatial stability across time (Fig. 3l) rather than trial numbers (Fig. 3k), although we noted a brief period of decreased spatial stability just after the VR turned on or was paused (Fig. 3k–l). In contrast, the mean spatial stability of spatial cells significantly decreased between the baseline and ketamine epochs, as well as between the control and ketamine epochs (Fig. 3m). This ketamine associated decrease in spatial stability was accompanied by a decrease in the mean spatial crest factor (a measure of how peaked fluctuations were in the spatial firing map, Fig. 3i, see "Methods") and a decrease in spatial information (Fig. 3j, "Methods"). Thus, ketamine affected firing rates heterogeneously across MEC cell types, increasing the firing rates of excitatory neurons while having little effect on the firing rates of interneurons, and broadly impaired spatial coding in the MEC.

To examine how functionally-defined excitatory MEC cell types responded to ketamine, we leveraged a previously published method for identifying grid cells in 1D linear track virtual reality environments ("Methods")[46,47]. As in the larger excitatory neuron population, the mean firing rate increased by ~2.5 Hz in putative grid cells after the ketamine injection, while the control injection had no effect on grid cell firing rates (Supplementary Fig. 7a). As in the larger spatial cell population, grid cell spatial selectivity degraded following ketamine injection (Supplementary Fig. 7b–e). However, unlike in the larger spatial cell population, grid cells did not recover spatial stability during the ketamine epoch (Supplementary Fig. 7d). Finally, we found that grid field widths did not change following ketamine administration (Supplementary Fig. 7f–g), suggesting the reduced spatial selectivity was not due to an increase in grid scale.

**Ketamine induced an acute discrete neural activity state characterized by reduced spatial information cell-pair co-activation**
To examine the effects of ketamine on MEC neural population activity, we applied an unbiased approach to classify temporal neural activity states. We first estimated the instantaneous firing rates of neurons by smoothing the vector of spike counts across temporal bins with a

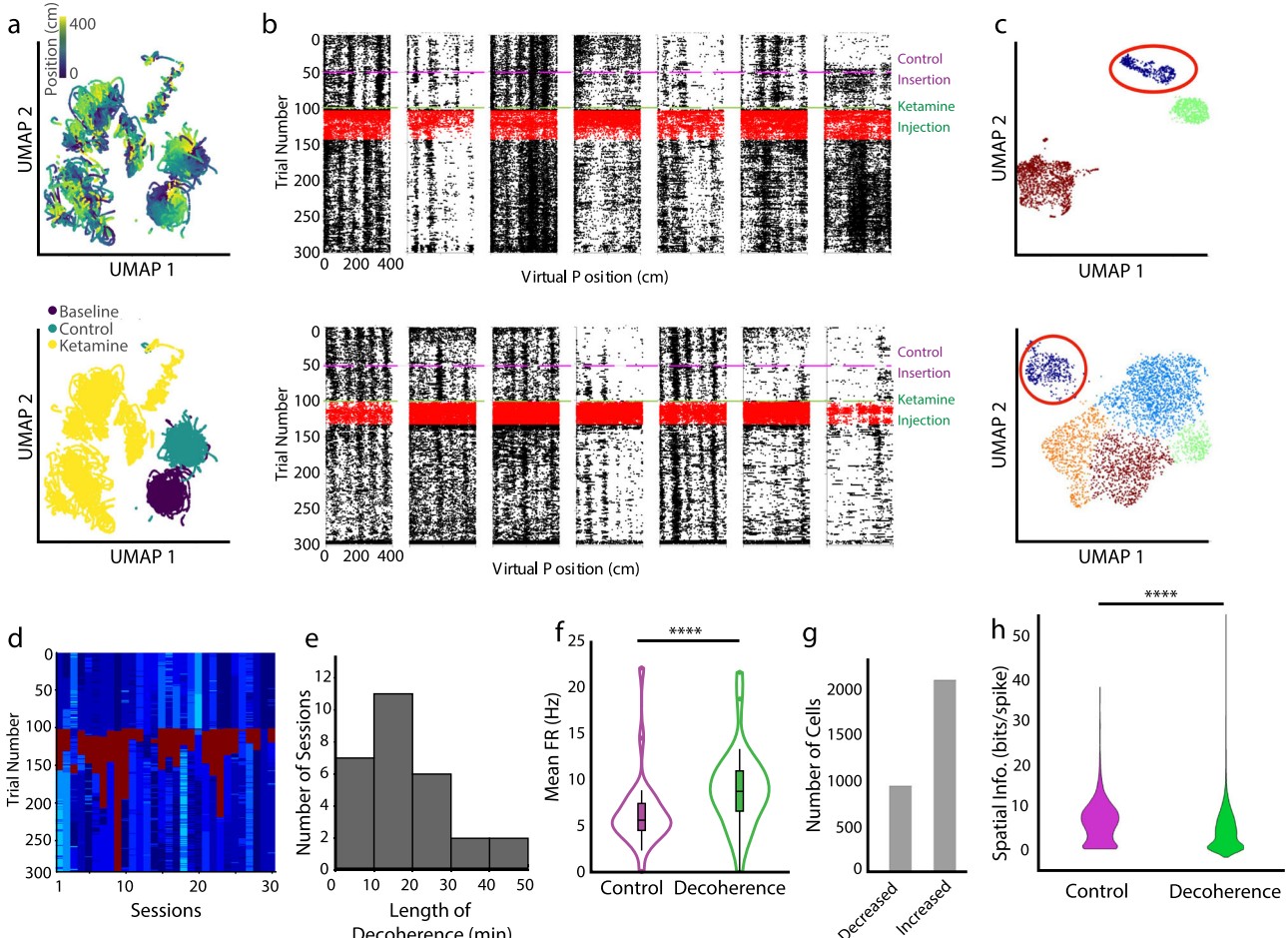

**Fig. 4 | Unbiased identification of the acute decoherence period evoked by ketamine. a** Temporally-binned firing rates plotted onto the first two dimensions of UMAP space. Each point represents a time-bin of population firing rates of recorded neurons. Color-coded by the animals position (top) and by the experimental epoch (bottom). Non-linear dimensionality reduction better clusters out activity by epoch and it is possible to visualize the animal's position on the looped-linear during the baseline and control sessions. **b** Raster plots from example cells with spikes identified in the acute-decoherence periods identified by UMAP and HDBSCAN, highlighted in red. **c** Two example sessions with firing rates dimensionally reduced with UMAP and clustered with HDBSCAN. Colors label clusters identified by UMAP and HDBSCAN. Circled cluster is the acute decoherence period that follows the ketamine injection. **d** UMAP/HDBSCAN identified trial clusters for each session, decoherence period highlighted in red. **e** A histogram of the length of identified acute decoherence periods (*n* = 28 sessions, 10 min bins). **f** Violin plot comparing firing rate in the decoherence period and the equivalently lengthened

control period, averaged over sessions. Firing rates are higher during the decoherence period (two-sided Wilcoxon matched pairs signed rank test, $Z = 22.78$, $p < 10^{-20}$, $n = 2926$ cells). The central mark of the boxplot indicates the median, the bottom and top edges of the box indicate the 25th and 75th percentiles, respectively, and the whiskers extend to the most extreme data points not considered outliers. **g** Histogram of the number of cells that decreased (left) or increased (right) their firing rate during the identified decoherence period ($n = 2895$ cells, 2007 cells increased, 888 cells decreased). **h** Violin plot comparing spatial information content in the decoherence period and an equivalent length control period. Spatial information is lower during the decoherence period (two-sided Wilcoxon matched pairs signed rank test, $Z = 16.5$, $p < 10^{-20}$, $n = 2926$ cells). Significant comparisons highlighted ****$p < 0.0001$. All violins have the same area, but the width represents the kernel probability density of the data at different values. Source data are provided as a Source data file.

Gaussian kernel. We applied a non-linear dimensionality reduction (Uniform Manifold Approximation and Projection: UMAP) and a clustering algorithm (Density-based spatial clustering of applications with noise: DBSCAN) to the neural firing rates, which revealed distinct clusters of population activity (Fig. 4a and Supplementary Fig. 8). We then focused on the first significant cluster of activity following the administration of ketamine (Fig. 4b, c. The well-defined activity cluster mapped back onto the period of spatial decoherence seen in the spatially-binned rasters, and we thus defined this as the "spatial decoherence period" (Fig. 4b, c). In 28 of 30 sessions, we observed an acute spatial decoherence period within 2 min of the administration of ketamine (Fig. 4d). The decoherence period was associated with an increase in mean firing rate compared to baseline and control epochs and lasted on average 18.6 ± 2.4 min (Fig. 4e–f). However, we also noted heterogeneity across individual cells during the decoherence

period. While there were, on average, more cells that increased their firing rate during the decoherence period, some cells decreased their firing rate (2.26 cells increased their firing rate for each cell that decreased their firing rate, Fig. 4g). Notably, the spatial information of cells was significantly lowered during the decoherence period (Fig. 4h).

Given ketamine's disruption of spatial information in MEC, we next considered whether ketamine impacted a cardinal feature of MEC spatial coding. Previous works point to MEC grid cell firing patterns as arising from attractor dynamics that emerge as a result of recurrent connectivity within a network of neurons[29,48]. A key feature of this framework is that pairs of co-active grid cells maintain their temporal firing relationship across environments and experimental conditions, even if their spatial firing relationships are disrupted (Fig. 5a). Indeed, cell-cell pairs have been found to retain their temporal firing relationships across a range of experimental conditions[30,49,50]. Given this,

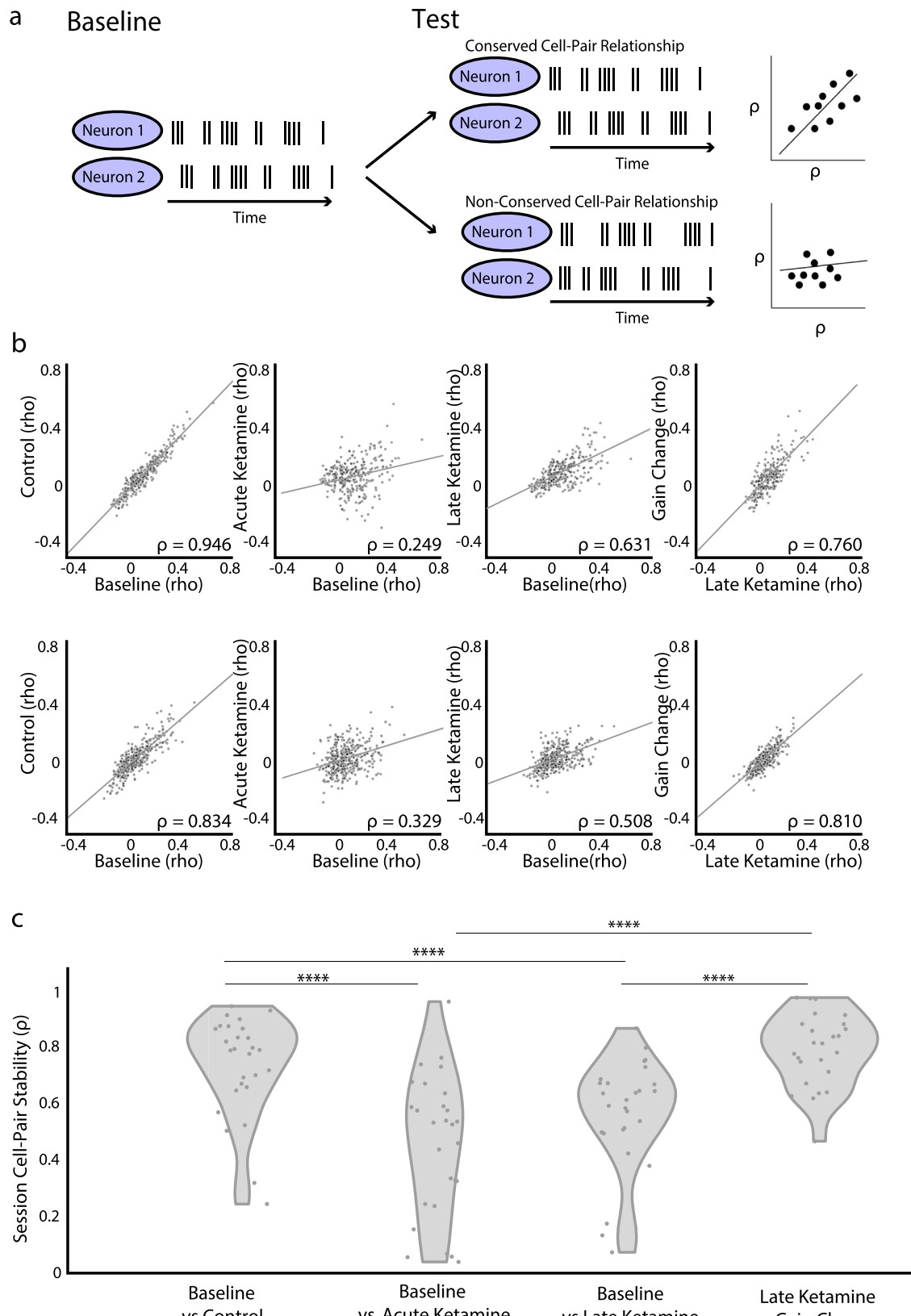

we examined whether cell-pairs maintain their temporal firing relationships across our different experimental conditions.

Consistent with previous work, we found that the temporal relationships between cell-pairs were maintained between the baseline and control conditions (Fig. 5b), as well as between late ketamine and a manipulation of the relationship between the virtual reality visual flow and locomotion (i.e. a change in the gain of the visual flow; Methods). In contrast, cell-pair stability was significantly diminished between baseline and acute ketamine conditions (Fig. 5c). This suggests that ketamine acutely disrupted cell-pair connectivity. In an attractor network framework, such a disruption in cell-cell co-activity would lead to a significant disorganization in stable spatial coding (i.e., the spatial

**Fig. 5 | Ketamine acutely disrupts cell-pair connectivity. a** Left: Schematic of a spike raster of a stable cell-pair relationship in the baseline epoch. Right: Spiking correlations between stable cell-pairs are conserved across perturbations and result in a diagonal relationship between spiking correlation coefficients (ρ) across two conditions (top right schematic). If a test perturbation breaks the relationship between two neurons, then that results in a non-diagonal relationship between the spiking correlation coefficients (bottom right schematic). **b** Two example sessions (each row is a session) of cell-pair correlations across different experimental epochs (baseline: trials 1–50, control: trials 51–100, acute ketamine: trials 101–150, late ketamine: trials 251–290, gain change: trials 291–300). To perturb the relationship between visual cues and locomotion, mice ran 10 trials where the gain was 0.5×. We calculated the correlation coefficient of between the smoothed firing rates of the first 50 baseline trials for all pairs of spatially stable (non-interneurons, spatial score >0.2) cells in a session, then examined cell-pairs with a significant correlation (Pearson correlation coefficient, $p < 0.05$). Each point is a cell-pair; line is the least squares line of the cell pairs. The Pearson correlation (ρ) value for each condition is at the bottom. **c** Violin plots of cell-pair correlations across different experimental states. All violins have the same area, but the width represents the kernel probability density of the data at different values. Each point (gray) indicates the Pearson correlation (calculated as shown in (**b**)) of a session. Sessions with >5 stable cell pairs are plotted ($n = 27$ sessions). Two-sided Wilcoxon matched pairs signed rank test with Sidak correction for multiple comparisons. Control vs acute ketamine: $Z = -4.31$, $p = 0.0001$; control vs late ketamine: $Z = -4.72$, $p = 1.4 \times 10^{-5}$; control vs gain change: $Z = 1.02$, $p = 0.89$; acute vs late ketamine: $Z = 1.49$, $p = 0.58$; acute ketamine vs gain change: $Z = 4.35$, $p = 8.2 \times 10^{-5}$; late ketamine vs gain change: $Z = 4.51$, $p = 3.9 \times 10^{-5}$). Significant comparisons highlighted ****$p < 0.001$. Source data are provided as a Source data file.

position a neuron is most active), which is consistent with our observation that ketamine disrupts spatial information in MEC.

## MEC spatial neurons remap following the discrete ketamine-induced neural activity state

Given that ketamine disrupts both the spatial information and temporal relationships between cell-pairs, we next considered whether the spatial representations observed before ketamine administration returned after the acute decoherence period. First, we noted that after the acute decoherence period, the spatial locations in which individual spatial neurons were most active differed from those observed in the baseline condition. The phenomenon of spatial neurons moving the spatial location they are most active in is often referred to as "remapping"[20,21] and is proposed to provide a population code for distinct environments[20–22] (Fig. 6a). To visualize this, we constructed trial-by-trial population spatial similarity matrices, in which the activity of all neurons for a given trial was correlated to their activity on all other trials[47] (Fig. 6b, c). We found that the population activity was spatially stable between the baseline and control period (trials 1–100). This was followed by a period of internal stability during the acute decoherence period. Of note, the trials following the ketamine administration were not spatially correlated with the baseline and control trials. To quantify the post-decoherence remapping of spatial firing patterns, we compared the spatial firing fields of neurons to a spatial template generated from the spatial firing fields of neurons in the baseline condition. Ketamine significantly decreased the correlation score as compared to the control injection (Fig. 6d–g). These results, combined with our previous analyses, reveal that ketamine induced an acute decoherence period in which the firing rates of MEC neurons and navigational behavior were significantly altered, followed by a return to stable, albeit remapped, spatial firing patterns.

Given the ketamine-induced changes to the firing rates and spatial coding of MEC neurons, we considered how these changes impacted the degree to which spatial information was preserved or disrupted across ketamine-induced neural activity states. We first trained a logistic regression decoder on data from the first 50 baseline trials (Methods, Fig. 6h–j). This decoder was significantly worse at predicting the animal's position on the VR track during the acute ketamine period compared to the control period (Fig. 6j). Together with the previous analyses, these results demonstrate that ketamine administration severely disrupts the ability of MEC to encode the spatial position of the animal.

## Ketamine disrupts spatial firing patterns in the hippocampus

The MEC is highly reciprocally connected with the hippocampus, which contains neurons called place cells that fire in one or few restricted spatial locations. Inactivating the hippocampus degrades grid cell firing patterns[32], and altering MEC activity impacts place cell firing patterns[51–53]. Thus, given the acute disruption caused by ketamine to neural activity in MEC, in a final set of experiments, we asked how ketamine influences the spatial firing patterns of place cells in the hippocampus. To examine ketamine's effect across a large population of hippocampal neurons, we performed in vivo 1-photon calcium imaging of CA1 neurons using a miniaturized fluorescence microscope ($n = 8$ mice, 4146 neurons; Fig. 7a). Mice freely explored a circular arena for one 15 min session across four days (days 1–2 served as baseline sessions, on day 3 ketamine was administered immediately before the start of the session and day 4 served as a post-ketamine baseline session; Fig. 7b). While the timescale of hippocampal experiments differed from those performed in MEC, due to constraints on the amount of time imaging can be performed, this approach allowed us to examine the impact of ketamine on the coding of large populations of hippocampal neurons within individual animals.

We observed that the mean and peak calcium event rate and information score were not significantly different between baseline sessions 1 and 2 (4146 cells from 8 mice). However, there was a significant decrease in mean and peak calcium event rate and information score between the baseline (day 2) and ketamine (day 3) sessions (Fig. 7c–e). The peak calcium event rate and information score returned to baseline values on the post-ketamine baseline session (day 4), but the decrease in mean calcium event rate persisted (Fig. 7c–e). This indicates that ketamine broadly decreased the activity of hippocampal CA1 neurons. We next considered how ketamine impacted CA1 place cells. First, we found that ketamine significantly decreased the number of cells defined as place cells in all 8 recorded mice (Fig. 7f). In addition, for the place cells identified in the baseline condition (i.e., baseline 2), we followed their activity in the ketamine session and observed significantly different spatial firing patterns between the two conditions (Fig. 7g). This loss of spatial tuning was not due to a complete shutdown of the cell activity (Fig. 7c), suggesting the activity of place cells was dissociated from the current location of the animal after the administration of ketamine, potentially in a manner similar to that observed in MEC during the acute decoherence period. Unlike the spatial cells in MEC, we did not observe place field remapping in the post-baseline after ketamine administration, as we observed similar spatial correlation values between the baseline and post-ketamine sessions as those observed between the two baseline sessions (Fig. 7h, i). One potential reason for this difference however, is that the post-baseline session was performed 1 day after the ketamine administration in the hippocampal experiments, compared to 30 min after ketamine administration in the MEC experiments. The possibility remains that over longer time scales (e.g., days), MEC neurons may return to their pre-ketamine baseline spatial firing patterns.

## Discussion

Previous clinical work has shown that, while ketamine is both a promising rapid treatment for depression, it can evoke dissociation (i.e., out-of-body experiences) and impair spatial memory[7,11,14,54]. Here, we examined ketamine-induced alterations in spatial memory at the level of behavior, single cells, and neural populations in two interconnected

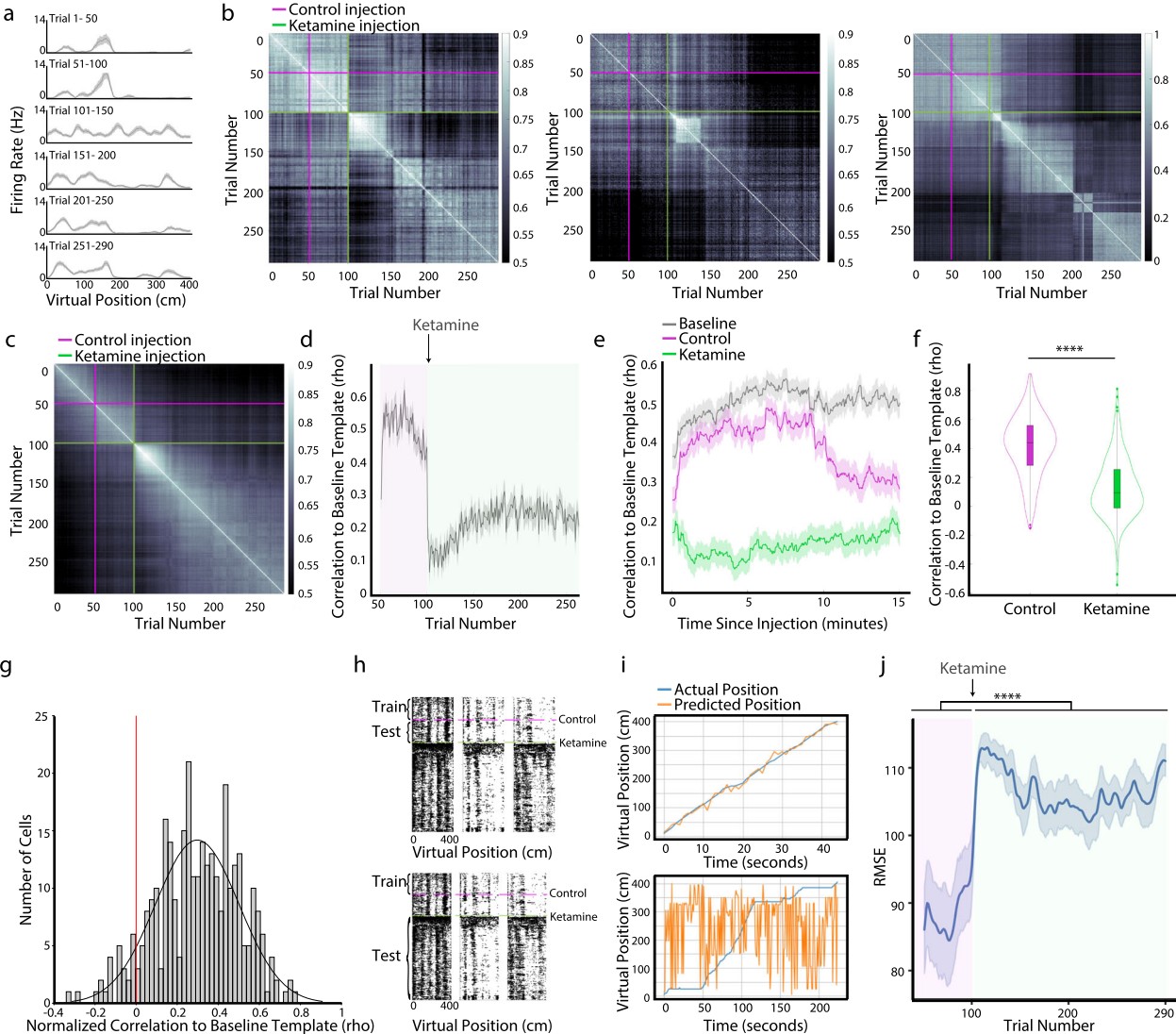

**Fig. 6 | Ketamine induces a long-term remapping following the acute decoherence period. a** Spatial tuning curve of an example cell split into 50 trial blocks. Solid lines represent mean and shaded regions represent SEM. **b** Trial-by-trial correlation matrices of spatial firing rates for three example sessions. Color indicates Pearson correlation between trials. **c** Spatial correlation matrix averaged across 30 recorded sessions (8 mice). **d** Correlation of each trial to the baseline spatial template (calculated from the 50 baseline trials, $n = 3233$ cells). Control epoch in magenta, and ketamine epoch in green. Solid lines represent mean and shaded regions represent SEM. **e** Correlation to the baseline spatial template for the first 15 min in the baseline (gray), control (magenta), and ketamine (green) conditions. Solid lines represent mean and shaded regions represent SEM. **f** Ketamine reduced spatial correlation to the baseline spatial template (two-sided Wilcoxon matched pairs signed rank test, $Z = 35.35$, $p < 10^{-10}$, $n = 3233$ cells). Width represents the kernel probability density of the data at different values. The central mark of the

boxplot indicates the median, the bottom and top edges of the box indicate the 25th and 75th percentiles, respectively, and the whiskers extend to the most extreme data points not considered outliers. **g** Distribution of difference between mean correlation of 25 trials preceding ketamine and mean correlation 25 trials following ketamine. **h** The logistic decoder was trained on spatially-binned firing rates from the 50 baseline trials, then tested on the control or ketamine epochs. **i** Example mouse position predictions from a control epoch trial (top) and a ketamine epoch trial (bottom). **j** The average root mean squared error (RMSE) of the logistic decoder is shown across all 30 sessions. Control epoch in magenta, ketamine epoch in green. Solid line is mean RMSE and shaded region is SEM. RMSE is higher following ketamine injection (two-sided Wilcoxon matched pairs signed rank test, $Z = -4.64$, $p = 3.51 \times 10^{-6}$, $n = 30$ sessions). Significant comparisons highlighted ****$p < 0.001$. Source data are provided as a Source data file.

regions involved in spatial navigation. In MEC, we found that ketamine altered network-level oscillatory dynamics (i.e., theta and gamma), acutely disrupted spatial coding by increasing firing rates and degrading the temporal firing-rate relationship between cell pairs and caused a longer-term remapping of spatial representations. Complementary to these observations in MEC, hippocampal spatial coding was disrupted during the same period of time as the acute disruption of MEC spatial coding. Taken together, our results point to the MEC-hippocampal circuit as a potential substrate for the effects of ketamine on spatial cognition.

The molecular mechanism by which ketamine affects spatial coding remains to be determined. Ketamine is a non-competitive *N*-methyl-D-aspartate (NMDA) receptor antagonist[55,56]. In MEC, NMDA receptor antagonists cause non-specific increases in excitation, as measured by c-fos mRNA and protein expression[23]. Ketamine is also a potent inhibitor of HCN1 ion channels and the associated ionic current I(h), with knockout of HCN1 channels reducing the hypnotic and dissociative actions of ketamine[16,57,58]. The loss of HCN1 channels and inhibition of I(h) impacts multiple features of MEC neural activity. At the network level, knockout of HCN1 channels decreases MEC

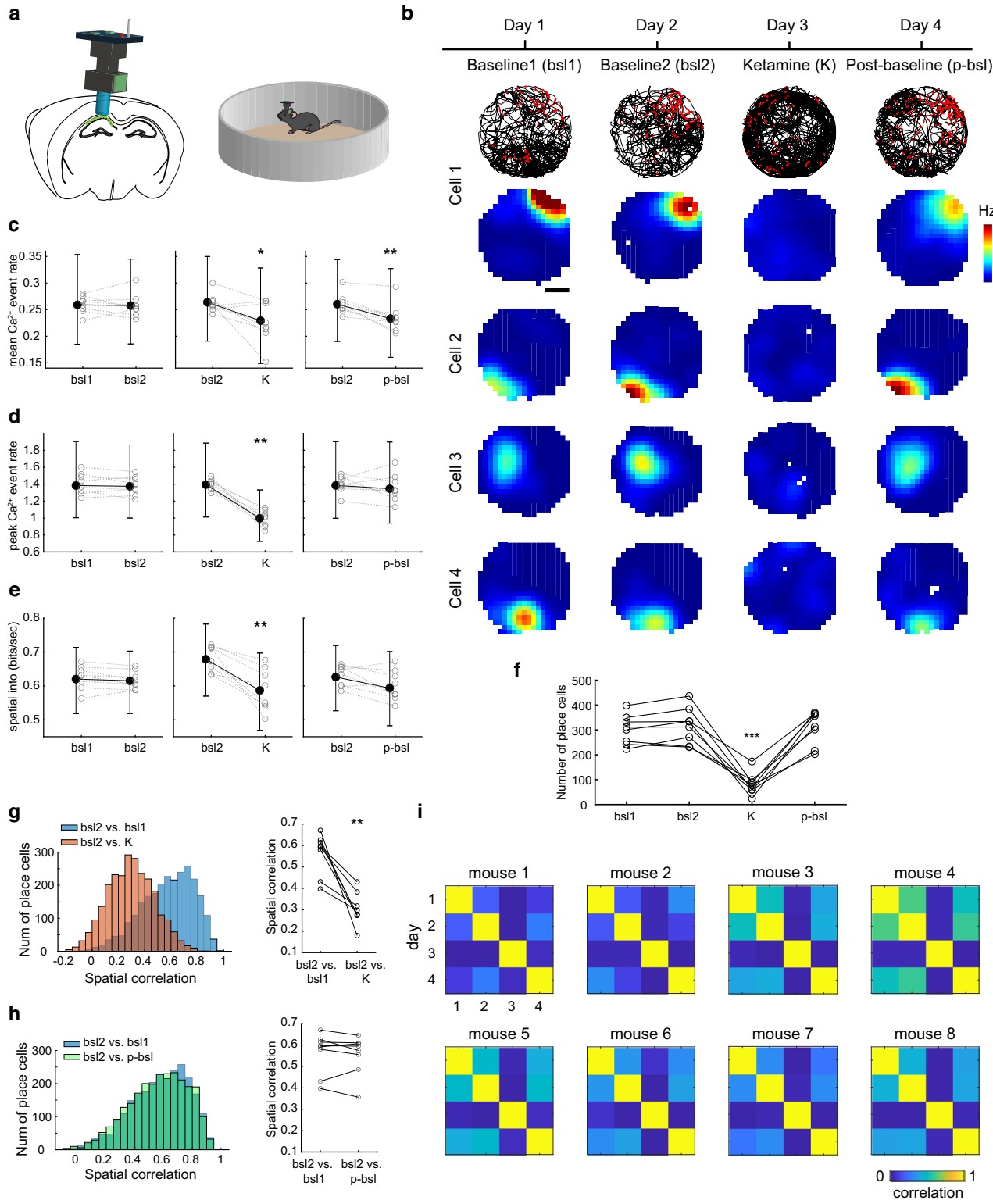

measured oscillations in the theta frequency band, consistent with our observation of decreased MEC theta after ketamine administration[59,60]. At the single cell level, knockout of HCN1 reduces the membrane resting potential, increases input resistance and enhances excitatory post synaptic potentiation summation[59,61,62]. Complementary to this, ketamine mirrors these single cell effects in MEC neurons in wildtype mice but not in HCN1 knockout mice[57]. These HCN1 associated effects of ketamine, likely together with ketamine's antagonistic effects on NMDA receptors, could contribute to the enhanced excitation

observed during the acute decoherence period and the longer-term remapping of MEC spatial representations. However, ketamine's effect on HCN1 has been mainly tested using higher, anesthetic doses[57,63,64]. Future work that combines genetic approaches with large-scale recordings will be required to pinpoint the potential contributions of HCN1 channels versus NMDA channels to ketamine-induced effects on MEC spatial coding.

Ketamine has different effects on physiology and cognition at different doses. In rodents, the IP anesthetic dose range of ketamine is

**Fig. 7 | Ketamine's effect on the spatial coding of hippocampal place cells. a** A schematic illustration of in vivo calcium imaging via a miniaturized fluorescence microscope in CA1. Reproduced with permission from Sun and Giocomo[96]. **b** Calcium event rate map of 4 example CA1 cells from 4 mice tracked across the experimental timeline shown on the top. Top, an example raster plot for one cell, where black lines are the animal's trajectory and red dots are the location of each calcium event. Scale bar = 10 cm. **c** Mean calcium event rate comparison on baseline 1 (bsl 1) vs. baseline 2 (bsl 2), bsl 2 vs. ketamine (K), and bsl 2 vs. post-baseline (p-bsl) from all recorded neurons (4146 cells from 8 mice). Open circles are means across animals, filled dot is median across animals, and error bars are 25th and 75th percentiles across neurons. Ketamine decreases the mean calcium event rate both during (K) and after (p-bsl) drug treatment ($p = 0.039$ and 0.0078, respectively, two-tailed sign-rank tested on 8 mice). **d, e** Organized as in (**c**). Ketamine decreases

the peak calcium event rate and spatial information of CA1 neurons ($p = 0.0078$ and 0.0078, respectively, two-tailed sign-rank tested on 8 mice). **f** Ketamine decreases the number of cells that qualified as place cell only during the drug day ($p = 9.51 \times 10^{-5}$, two-tailed paired $t$ test). Each dot is one mouse. **g** Left, histogram of spatial correlations between sessions (2536 place cells from 8 mice). Right, spatial correlation per animal averaged over cells. Spatial correlation declines on bsl 2 vs. K compared to the correlation from the two baseline days ($p = 0.0078$, two-tailed sign-rank test on 8 mice). **h** Organized as in (**g**). There is no significant difference between the spatial correlation of bsl 2 vs. bsl 1 and bsl 2 vs. p-bsl ($p = 0.38$, two-tailed sign-rank test on 8 mice). **i** Spatial correlations averaged over all place cells per day. Ketamine consistently decreases spatial correlations compared to baseline sessions. Significant comparisons highlighted *$p < 0.05$, **$p < 0.01$, and ***$p < 0.001$. Source data are provided as a Source data file.

150+ mg/kg[65,66] and the sub-anesthetic sensory dissociative range is 25–100 mg/kg[16,18]. However, spatial memory impairments have been observed in doses as low as 2.5 mg/kg[67]. The dosage effects may be in part due to different affinity for different target receptors. For instance, ketamine's affinity for HCN1 is much lower than its affinity for NMDAR[56,57]. Here, we find changes in spatial navigation and spatial representations at a low dissociative dose. Future work should examine neural activity changes in vivo at lower doses of ketamine, particularly at subanesthetic non-dissociative doses.

Interestingly, ketamine increased firing rates in MEC while decreasing calcium event rates in CA1. Ketamine has been shown to have different effects across different brain regions[68]. Several ex vivo studies have found that ketamine acutely reduces excitability in CA1[25,68–70] (but see also Widman and McMahon[71]). In contrast, in vivo studies have found that ketamine acutely increases excitability in MEC[23]. Through its antagonism of NMDAR on inhibitory neurons, ketamine reduces activity of inhibitory neurons in ex vivo hippocampal slices, disinhibiting excitatory neurons[72–74]. Ketamine therefore likely reduces CA1 excitatory neuronal activity through another mechanism, such as HCN1 channels. Our results also suggest that the increased firing rate of MEC excitatory neurons was not due to disinhibition by blocking NMDAR on inhibitory neurons, but instead increased effectiveness of excitatory synaptic transmission potentially combined with increased intrinsic excitability or increased external inputs.

Another possible mechanism by which ketamine administration might alter MEC spatial coding is via its effects on the hippocampus. Importantly, back-projections from the hippocampus to MEC are important for grid patterns in the MEC, as inactivating the hippocampus deteriorates grid cell firing patterns[32]. A role for intact hippocampal input in the emergence of MEC spatial coding is consistent with attractor-network models, a class of computational models capable of generating grid cell firing patterns[29,33,75]. In these models, grid cell firing patterns emerge from a network of neurons with asymmetrically centered inhibitory center-surround synaptic weight profiles. This connectivity profile, in the presence of broad-field excitation, will generate a grid pattern of activity. Individual grid cell responses then emerge when the movement of the pattern of activity across the network is coupled to the velocity of the animal. If the inhibitory neurons compose the inhibitory center-surround synaptic weight profiles of the grid network, a reduction in the excitatory drive from the hippocampus can result in a loss of correlations between grid cell pairs[30,32]. Our finding that hippocampal place cell activity diminishes during the acute ketamine period, combined with the decrease in cell-pair temporal correlations of MEC cells during the same period, is consistent with this framework and points to the degradation in hippocampal place cells as a possible circuit mechanism for ketamine-induced changes in MEC spatial representations. However, any circuit mechanism likely interacts with ketamine's effects on MEC NMDA receptors or HCN1 channels. For example, previous work has shown that inactivation of the hippocampus does not result in an increase in

firing rates of MEC neurons[32]. Thus, the effect of ketamine on MEC spatial representations likely reflects an interplay between ketamine-induced circuit, cellular and molecular changes.

Ketamine may also affect spatial coding through its effect on theta and gamma frequency oscillations. Theta frequency oscillations organize spatial cell firing in CA1[76,77] and MEC[38,39]. The reduced theta power we observed cannot explain the spatial decoherence, as theta oscillations do not affect spatial firing of grid cells[78–80], but could explain the loss of spatial stability, as reducing theta power in MEC by silencing the medial septum (MS) impairs MEC spatial stability[41]. Specifically, this spatial stability was lost in grid cells and not non-grid spatial cells, which could explain why we observed that spatial stability recovered in excitatory cells following acute ketamine, but not in grid cells. MS also projects to CA1 to drive theta oscillations, but whether suppressing theta by silencing MS reduces the spatial stability of place cells depends on the method of inactivation used[41,81]. This mechanism of altering spatial navigation circuits could be through ketamine's effects on GABAergic neurons in MS. Suppressing theta by silencing GABAergic neurons in MS disrupts grid stability, but suppressing theta by silencing cholinergic neurons does not[80]. Moreover, ketamine's effect on gamma oscillations is suppressed by silencing GABAergic, but not cholinergic, MS neurons[37]. Fast gamma in particular coordinates communication between MEC and CA1[82]. The observed increase in fast gamma could lead to aberrant synchrony between MEC and CA1, similar to the aberrant synchrony between cortical regions observed during ketamine-induced sensory dissociation[16]. Thus, both the decrease in theta power and increase in fast gamma power we observed may be through ketamine's effect on MS and could impair spatial representations.

The effects of ketamine on MEC and hippocampal spatial representations likely interact with the effect of ketamine on other regions involved in spatial navigation, such as the retrosplenial cortex[16]. Spatial navigation and its supporting cortical and subcortical regions also receive direct and indirect inputs from visual and somatosensory cortices. Alterations in spatial representations may be in part due to altered primary cortex representations that lead to the visual and proprioceptive hallucinations found at subanesthetic dissociative doses[2,83]. Given that ketamine produces a wide repertoire of cognitive changes, ranging from unconsciousness, dissociation and spatial memory impairments to clinically relevant improvements in depression and anxiety, future work will be needed to disentangle how ketamine's effects on specific brain circuits are associated with specific cognitive changes. For example, infusing ketamine into MS could reveal whether the loss of spatial stability and temporal relationships between pairs of cells we observed was due to decreased theta from MS to MEC. Furthermore, with the advent of technologies that can administer drugs with higher anatomical specificity[84], for example, it may soon be possible to apply ketamine in a way that targets depression without inducing disturbances in spatial cognition. Another consideration for future work is how ketamine impacts neural activity in freely moving animals. Here, we chose to leverage the reduced

freedom of movement that occurs in a 1D head fixed condition, as it allowed better comparison of spatial maps after ketamine since the mouse must run along the same path. In freely moving 2D environments, spatial cells are nearly impossible to observe, as ketamine induces a stereotyped spinning behavior (Supplementary Fig. 3e). Thus, the challenge remains of how to best consider the spatial coding of neurons after ketamine administration in freely moving animals.

## Methods

### Subjects for electrophysiological recordings

All techniques were approved by the Institutional Animal Care and Use Committee at Stanford University School of Medicine. Neural recordings with ketamine injections were made from 8 mice (5 wild type C57Bl/6 and 3 wild type 129S6/SvEvTac) aged 12 weeks to 16 weeks at the time of the first surgery (14.3–24.4 g). All mice were female except for 2 mice. An additional 2 mice (female wild-type 129S6/SvEvTac) received two blank IP injections (control insertion); data from these mice are only shown in Fig. 1e. Animals were housed at 21–23 °C and 30–70% humidity. Before surgery, mice were group housed with same-sex littermates, unless separation was required due to water restriction or aggression. After surgery, mice were housed individually. Mice were housed in transparent cages on a 12-h light-dark cycle and experiments were performed during the light phase. All data were collected by male experimenters, which has been shown to have an effect on CA1-projecting entorhinal cortex corticotrophin releasing factor neurons[85].

### In vivo survival surgeries

For all surgeries, anesthesia was induced with isoflurane (4%; maintained at 0.5–1.5%) followed by an injection of buprenorphine (0.05–0.1 mg/kg). Animals were injected with baytril (10 mg/kg) and rimadyl (5 mg/kg) immediately following both the first and second surgery and for 3 days afterwards. In the first surgery, animals were implanted with a custom-built metal headbar containing two holes for head fixation. The craniotomy sites were exposed and marked during headbar implantation and the surface of the skull was coated in metabond. Additionally, a jewelers' screw with an attached gold pin, to be used as a ground, was implanted anterior to the metal headbar. After completion of training, a second surgery was performed to make bilateral craniotomies at 3.3 mm lateral to the central suture and immediately anterior of the transverse sinus. A small plastic well was implanted around each craniotomy and affixed with metabond. Craniotomy sites were covered with a drop of sterile saline and with silicone elastomer (Kwik-sil, WPI) in between surgery and recordings.

### Virtual reality (VR) environment

The VR recording set-up was based on that described in Cambell et al.[86] and Low et al.[47]. Head-fixed mice ran on a 15.2-cm-diameter foam roller (ethylene vinyl acetate) constrained to rotate about one axis. The cylinder's rotation was measured by a high-resolution quadrature encoder (Yumo, 1024P/R) and processed by a microcontroller (Arduino UNO) using custom software.. The virtual environment was displayed on three 24-inch monitors surrounding the mouse and generated using commercial software (Unity 3D) and updated according to the motion signal. The gain of the virtual reality track was calibrated so that the virtual track was 400 cm long. Upon completing the track, the mouse was teleported seamlessly back to the start in a way such that the track seemed infinite to the mouse (visual cues of the next track were repeated and visible in the distance as the animal approached track end). The floor of the virtual environment was a checkerboard texture, and there were 5 pairs of evenly spaced (80 cm apart) visual towers that were the same on the left and right sides of the track (see Fig. 1b). Each of the 5 pairs of towers were different heights, widths, and patterns (all were black and white and had neutral luminance). Water rewards were delivered via Tygon tubing attached to a metal lick spout mounted in front of the mouse, and delivery was triggered via a solenoid valve, which produced an audible clicking sound upon reward delivery. The mouse's licks were detected with a custom-built infrared light barrier. The Neuropixels probes were mounted on a motorized micromanipulator (UMP Micromanipulator, Sensapex). Probe holders were placed behind the mouse to minimize visual disturbance.

### Recording session structure and drug administration

Mice ran 300 trials per recording session. The first 50 trials served as the baseline epoch, in which no manipulation was performed. Trials 51–100 served as the control epoch, which occurred after a blank IP injection (control insertion) was given. Trials 101–290 served as the ketamine epoch which occurred after the administration of a 25 mg/kg IP injection. Trials 290–300 served as the gain manipulation trials where the gain of visual flow was set to 0.5 speed relative to the animal's movement on the running wheel. IP injections were delivered to the head-fixed animals in about 10 s between VR trials. The 25 mg/kg dose of ketamine was determined based on previous literature and an in-house ketamine dose-response curve (Supplementary Fig. 3 and see "Ketamine dose response curve" section of "Methods").

### Training and handling

After headbar implantation, mice recovered for three days and then were put on water deprivation. They received 0.8–1.0 mL of water each day and their weights were monitored to ensure that they remained above 80% of baseline. Mice were handled for 15 min at least once every 2 days following headbar implantation and given an in-cage running wheel. Training progressed in three stages. In stage one, they were head-fixed on the VR rig and trained to receive water from the lickspout. Water delivery was associated with an audible click of the solenoid. Mice quickly learned this association and began licking upon hearing the click. After stage one, mice progressed to stage two, in which they ran on a training track consisting of a floor with checkerboard texture and evenly spaced, visual landmarks on both sides to receive water rewards at a reward tower. Water rewards (2 µL) were automatically delivered whenever the mouse passed the reward tower. The reward tower spacing started at 40 cm and the track was lengthened daily up to a maximum of 200 cm, such that the reward tower moved further down the track, to encourage running. The reward tower on the habituation track was visually identical to the reward tower on the track used for recording. Once mice ran consistently on the training track (average running speed >10 cm/s), they progressed to stage three, in which they ran on the same track that would eventually be used for recording, increasing from 50 to 400 trials per day. During this final phase of training, mice developed stereotyped running and licking patterns in which they slowed down and licked prior to the reward tower. Some training sessions were performed on a training rig, but mice were always trained on the recording rig for several days prior to the first recording day to familiarize them with the setup. Mice were deemed fully trained and ready to record when they completed 400 trials within 1 h for 2 consecutive days. Mice that never learned the task were excluded from further experiments.

### In vivo electrophysiological data collection

All recordings were performed at least 24-h after the craniotomy surgery, at which point the mouse was head-fixed on the VR recording rig. The craniotomy site was exposed and rinsed with saline; debris was removed using a syringe tip. Recordings were performed using Phase 3B Neuropixels 1.0 silicon probes[34] with 384 active recording sites (out of 960 sites total) located at the bottom ~4 mm of a ~10 mm shank (70 µm wide shank diameter, 24 µm thick, 20 µm electrode spacing), with the reference and ground shorted together. The probe was positioned over the craniotomy site at 10° from vertical and targeted to ~50–300 µm anterior of the transverse sinus using a micromanipulator. On consecutive recording days, probes were targeted

medial or lateral of previous recording sites, as permitted by the craniotomy. The reference electrode was then connected to a gold ground pin implanted in the skull. The probe was advanced slowly (-10 μm/s) into the brain until it encountered resistance or until activity quieted on channels near the probe tip, then retracted 100–500 μm and allowed to sit for at least 30 min prior to recording. While the probe was implanted, the craniotomy site was covered with sterile saline and silicone oil. Signals were sampled at 30 kHz with gain = 500 (2.34 μV/bit at 10 bit resolution) in the action potential band (high-pass 300+ Hz filter), digitized with a CMOS amplifier and multiplexer built into the electrode array, then written to disk using SpikeGLX software. Local field potentials were similarly sampled at 2.5 kHz with gain = 250 (4.69 μV/bit at 10 bit resolution) and 0–300 Hz low-pass filter.

## Histology and probe localization

In order to identify the probe location after the completion of all experiments, probes were dipped 10 times at 10 s intervals in a fixable lipophilic dye before each probe insertion (1 mM DiI, DiO or DiD, Thermo Fisher). Mice were sacrificed with an overdose of pentobarbital and transcardially perfused with phosphate-buffered saline (PBS) followed by 4% paraformaldehyde within 7 days of the first probe insertion. Brains were extracted and stored in 4% paraformaldehyde for at least 24 h before transfer to 30% sucrose in PBS. The brains were then flash frozen, cut into 45-μm sagittal sections with a cryostat, mounted and stained with cresyl violet. Histological sections were examined and the location of the probe tip and entry into the dorsal MEC for each recording were determined based on the reference Allen Brain Atlas[87] (Supplementary Fig. 1). The location of each recording site along the line delineated by the probe tip and entry point was then determined based on each site's distance from the probe tip. Depth reported is the ventral distance from the location of the dorsal boundary of MEC in the medial section where the probe enters MEC.

## Offline spike sorting

Neuropixels probes electrophysiological recordings using SpikeGLX software and Matlab were common-average referenced to the median across channels and high-pass filtered above 150 Hz. Automatic spike sorting was then performed using Kilosort2, a high-throughput spike sorting algorithm that identifies clusters in neural data and is designed to track small amounts of neural drift over time (open source software by Marius Pachitariu, Nick Steinmetz, and Jennifer Colonell, https://github.com/MouseLand/Kilosort2)[88]. All clusters with peak-to-peak amplitude over noise ratio <3 (with noise defined as the standard deviation of voltage traces in a 10 ms window preceding detected spike times), total number of spikes <100, and repeated refractory period violations (0–1 ms autocorrelogram bin >20% of maximum autocorrelation) were excluded after automatic spike-sorting. All remaining clusters were manually examined using Phy (an open-source Python graphical user interface for visualization and manual curation of large-scale electrophysiological data) and labeled as "good" (i.e. stable and likely belonging to a single, well-isolated neural unit), "MUA" (i.e. likely to represent multi-unit activity), or "noise." This paper only analyzed well-isolated "good" units from within MEC with greater than 100 spikes from sessions with >10 cells.

## Virtual reality behavioral data preprocessing

Virtual position and time stamps were recorded on each frame of the VR scene, and a synchronizing TTL pulse was generated from an Arduino UNO and recorded in SpikeGLX using an auxiliary National Instruments data acquisition card (NI PXIe-6341 with NI BNC-2110). The time of each lick (identified by the breaking of an IR beam by the mouse tongue) was also synchronized to the neurophysiological data. Time stamps were adjusted to start at 0 and all behavioral data was interpolated to convert the variable VR frame rate to a constant frame rate of 50 Hz. Since the virtual 400 cm track was effectively infinite,

recorded positions less than 0 or greater than 400 cm were converted to the appropriate position on the circular track. Trial transitions were identified as timepoints where the difference in position across time bins was less than −100 cm (i.e. a transition from ~400 cm to ~0 cm) and a trial number was accordingly assigned to each time point. Running speed for each time point was computed by calculating the difference in position between that time point and the previous, divided by the framerate (speed at the first time point was assigned to be equal to that at the second time point). Speed was then interpolated to fill removed time points and smoothed with a Gaussian filter (standard deviation 0.2 time bins).

## Statistics

Analysis was conducted in Matlab and Python. All tests were two-sided unless otherwise noted, values are presented as mean ± standard error of the mean (SEM), and correlation coefficients are Pearson's correlations unless otherwise noted. Wilcoxon signed-rank tests and Wilcoxon rank-sum tests were used to assess significance for paired and unpaired data respectively. Experimenters were not blinded during data collection and experimental analysis. Sample sizes were consistent with previous similar studies and not predetermined.

## Lick accuracy and running speed

Licking behavior was measured with an infrared light barrier. Breaking the IR light barrier resulted in photodiode voltage output drops, which were monitored by an Arduino UNO. The VR computer queried the voltage on each frame and individual licks were defined as the voltage dropping below a predefined threshold. Accurate licks were defined as occurring within 50 cm of the reward tower placed at 400 cm. VR lick accuracy percentage was calculated by dividing the number of accurate licks by the total number of licks. Trials where the animal did not lick were considered to have a lick accuracy of 0. To calculate the running speed of the mouse, we calculated the difference in VR position between consecutive VR frames. The resulting running speed trace was smoothed with a Gaussian kernel ($\sigma = 0.2$ s).

## Local field potentials

Local field potentials were analyzed on one channel per session, selected as the channel within MEC with the highest theta power. Point source density was calculated using Welch's method from the scipy signal package. Power in each frequency band (5–11 Hz for theta, 50–110 Hz for fast gamma) was calculated using a multitaper spectrogram using the spectral_connectivity package[89] with a window of 1 s. Frequency band power traces were $Z$-scored to the mean and standard deviation of the baseline epoch. Frequency band power traces from the control and ketamine epochs were then resampled to match the velocities of the baseline epoch, to control for the effects of speed on theta power. For each 1 s time bin during the baseline epoch, a matching time bin was selected, with replacement, from the paired epoch that most closely matched the mean velocity in that time bin.

## Spatial information, stability, and crest factor score in VR

Spatial information was calculated in bits per second over 2 cm position bins[90]. Firing rates were computed empirically (number of spikes in position bin i divided by occupancy time). Spatial firing rate vectors were generated by dividing the linear VR track into 2 cm bins and dividing the number of spikes in each bin by the dwell time in that bin. Time periods when the mouse was moving at less than 2 cm/s were omitted from the analysis. A spatial stability score was calculated by smoothing the firing rate vector with a Gaussian filter (standard deviation = 2 cm), normalizing the data, correlating each trial to the preceding trial and the subsequent trial, and finding the mean between those two values. The stability of the first and last trial were calculated by correlating with only the subsequent and preceding trial respectively. "Crest factor" scores for each trial measuring the "peakiness" of

each cell's signal were calculated from a cell's spatial firing rate vectors by calculating the peak value divided by the root mean square of the spatial tuning curve, as shown in Eq. (1).

$$x_{crest} = \frac{x_{peak}}{\sqrt{\frac{1}{N}\sum_{i=1}^{N} x_i^2}} \tag{1}$$

## Transmission probability

Transmission probability for each identified excitatory and inhibitory connection were calculated as previously described[44,45]. First, a cross-correlogram for each cell pair (cell A to cell B) was generated. The spike times of the downstream neuron (cell B) were temporally shuffled 1000 times. A pair was considered a monosynaptic connection if the cross-correlogram values in the 5 ms following a spike from cell A went above (excitatory) or below (inhibitory) the 99% confidence interval generated by the shuffled data. Cross-correlograms were then generated for each 10 s time bin for each significant monosynaptic connection and divided by the number of spikes from cell A, so each cross-correlogram was a spiking probability given a single spike from cell A. Transmission probability was the area under the cross-correlogram curve at 1–5 ms after a spike from cell A minus was the area under the cross-correlogram curve at 1–5 ms prior to a spike from cell A, to correct for the effects of changes in the firing rate of cell B independent of cell A's influence. This value reflects the extent to which a spike from cell A increased firing of cell B independent of cell B's firing rate.

## Functional cell-type identification

Spatially stable cells were classified by first generating spatial firing rate vectors by dividing the linear VR track into 2 cm bins and dividing the number of spikes in each bin by the dwell time in that bin. Time periods when the mouse was moving at less than 2 cm/s were omitted from the analysis. To identify spatially stable cells at baseline, we examined the firing rate vectors for each neuron across the 50 baseline trials. The firing rate vectors were smoothed with a Gaussian filter (standard deviation = 2 cm), the data were normalized, and baseline spatial stability scores were generated calculating the mean spatial stability score (see above) for the baseline epoch. Cells with a baseline spatial stability >0.2 were classified as spatially stable cells.

Putative high-firing rate interneurons were identified by finding cells with a mean firing rate >15 Hz (putative high-firing interneurons). Excitatory neurons had mean firing rates <15 Hz. Animals ran 10 trials following the completion of the recording session (trials 291–300) where the correlation between the VR visual environment flow and locomotion was manipulated to have a gain of 0.5×, such that the animal had to travel two times the normal distance to cover the same amount of visual track. Previous work has shown disassociating the visual flow from locomotion with a gain <1 can be used to identify putative grid cells—as grid cells in MEC are more sensitive to gain change[46,86]. We computed the spatial tuning curves for the last 10 normal trials and the 10 gain manipulation trials and calculated the Pearson correlation between these tuning curves for each cell. Putative grid cells were identified as excitatory cells whose spatial tuning curves were modulated by changing the visual gain (rho < 0.2), whose spatial firing fields were stable across baseline trials (rho > 0.2), and whose spatial information score averaged across baseline trials was >3. Note this method is a rough classification of grid cells that likely includes some false positives and may falsely reject some true grid cells.

To calculate field widths for each grid cell, we took the firing rate over each position bin and averaged this spatial map over 50-trial blocks. We spatially shuffled each map 1000 times and set the detection threshold for peaks as the 80th percentile of all peak heights from these shuffles, similar to previous methods[46]. Field width was the maximum width of all detected peaks in each 50-trial block, measured at the level of half the height of the peak.

## Identifying the decoherence period using UMAP/DBSCAN

Temporally binned firing vectors for each recorded cell were generated by binning spikes into 0.02 s bins. The population of recorded cells per session was then stacked together to produce a matrix C of size (number of cells, number of time bins). UMAP and DBSCAN were performed using corresponding MATLAB packages[91–93]. UMAP dimensionality reduction was then performed on matrix C to get embedding coordinates for each cell. Parameters for UMAP for this embedding were n_neighbors = 15, n_components = 2, distance metric = euclidean, min_dist = 0.3. Clustering with DBSCAN was performed in the three-dimensional embedding space to identify clusters of neural activity corresponding to the main manifolds. The first major cluster of neural activity (duration >1 min), which started between 2–15 min after the ketamine administration, was labeled the "decoherence period."

## Cell-pair connectivity analysis

To calculate cell-pair connectivity, we first calculated the Pearson correlation coefficient between the smoothed temporally-binned firing rate of the first 50 baseline trials for all spatially stable pairs of cells in a session. We then identified cell-pairs as those with significant Pearson correlations ($p < 0.05$) during the baseline epoch. We then compared the correlation coefficients of these cell-pairs across five different epochs: baseline (trial 1–50), control (trials 51–100), acute ketamine (trials 101–150), late ketamine (trials 251–290), and the visual-gain change (291–300) epochs. During the 10 trial visual-gain change epoch, the correlation between the VR visual environment flow and locomotion was manipulated to have a gain of 0.5×, such that the animal had to travel two times the normal distance to cover the same amount of visual track. We chose to focus on gain values <1, as these tended to show more strongly the influence of path integration on spatial firing in MEC[46,86]. We calculated the least squares line of the cell-pairs in the different conditions and a Pearson correlation (ρ) value for each the following comparisons: baseline versus control, baseline versus acute ketamine, control versus late ketamine, late ketamine versus visual-gain change.

## Trial × trial population similarity matrices and correlation scores

Spatially binned firing matrices $A_i$ (trial × position) for each recorded cell i = 1...N from a single session were concatenated to form the population matrix $P = [A_1|A_2|...|A_N]$, where N is the number of cells recorded in that session. We computed the trial x trial population similarity matrix $S$ for each session by calculating the Pearson's correlation coefficient between each row (trial) in the population matrix $P$ ($S_{ij} = \text{corr}(P_i^T, P_j^T)$). To generate the averaged spatial correlation matrix $\underline{S}$, we found the mean of the 30 recorded session's trial × trial population similarity matrices.

To calculate the correlation score which quantified the post-decoherence remapping of spatial firing patterns, we first created a "baseline template" for each cell in a session by computing the mean spatial firing rate tuning curve across the baseline epoch (trials 1–50), denoted as $b$ ($b_i = \text{mean}(A_i [a_1^T...a_{50}^T]$, where $a_i^T$ is the i$^{th}$ row of $A_i$). We then calculated Pearson's correlation coefficient between the baseline spatial template $b$ and the remaining trials of the spatial firing fields (corr($b_i$, $A_i [a_j^T]$) for j = 51...290).

## Logistic decoder

We used a logistic regression model(penalty = "l2," random_state = 0, solver = "lbfgs," multi_class = "multinomial," max_iter = 10,000,000, C = 0.03) to predict the animal's position in VR from the spiking activity of all recorded neurons in a session, and we referred to the optimized model as a "decoder" following common terminology and practice[94]. We used the baseline epoch (trials 1–50) as the training (encoding) trials. The z-scored firing rate was used to compute spatial tuning curves for encoding trials by averaging the population activity within

each spatial bin (2 cm). This defined an average $N$-dimensional trial trajectory during encoding trials, where $N$ is the number of neurons. The model was trained 50 times dropping 1 to find the baseline RMSE. We then tested the model with the neural population activity from each subsequent trial during both the control epoch (trials 51–100) and ketamine epoch (trial 101–290). For each time point during the decoding (test) trials, we determined the closest point on the average encoding trajectory by minimizing the Euclidean distance. The decoded position then corresponded to the VR position for this point on the average trajectory. We calculated the decoding error as the root mean squared error (RMSE) between decoded position and actual position of the animal respecting the circularity of the track. We then found the mean RMSE curve by averaging the RMSE curves (trials 51–290) across the 30 recorded sessions ($n = 8$ mice).

### Ketamine dose–response curve and open-field behavior

To test spatial mobility, 5 female and 5 male (C57BL/6 mice, aged 12 weeks to 24 weeks) were allowed to freely explore a 60 cm × 60 cm open field. In order to minimize external visual cues, the open-field was surrounded by black walls. A single white cue was located on one wall to orient the animal. In order to minimize the effect of external noise, the open field arena was be placed in a 49"($l$) × 49"($w$) × 102"($h$) soundproof studio room (reduces noise above 1000 Hz by greater than 60 dB, and reduces noise from 250 Hz to 100 Hz by -50 dB). A HEPA filter was run constantly inside the behavioral box to both provide a constant source of white noise and to clean the air of particulates and odors. The open-field behavioral tasks were recorded via a ceiling mounted infrared camera. Etho-Vision XT was used to analyze the behavior including animal tracking, animal speed, and animal sniffs of the objects (Supplementary Fig. 3b–e).

To examine spatial memory consolidation, we tested 14 female mice (C57BL/6 mice, aged 12 weeks to 24 weeks) on the Object-Location Memory task (Supplementary Fig. 3a, b). This task specifically tested hippocampal-dependent spatial memory without needing an aversive stimulus[95]. This task leveraged the fact that mice naturally spend more time exploring novel objects and the fact that mice notice when objects have been moved to a new location. Mice were exposed to an open field arena with two distinct objects for 10 min. Half the mice ($n = 7$) were given 25 mg/kg ketamine and the other half ($n = 7$) were given a control injection. The following day, one of the objects was moved to a new spatial location and the mice were allowed to explore freely for 5 min. If the animal remembered the previous day, it would spend more time investigating the object that had moved. The animals were recorded using an infrared camera mounted above the arena, and the amount of time spent sniffing the objects was scored using EthoVision XT. The videos were analyzed and the discrimination index (DI = {[time spent exploring object in novel location – time spent exploring object in familiar location]/total time exploring both objects} during the test phase was calculated as a measure of spatial memory.

In order to test the dose-response curve of ketamine, mice were given ketamine at the following doses: 0 mg/kg, 10 mg/kg, 12.5 mg/kg, 15 mg/kg, 20 mg/kg, and 25 mg/kg once a day IP (Supplementary Fig. 3d). The drug was delivered to mice through an IP injection and then immediately placed in the open field arena to be tracked for 30 min afterwards using EthoVision XT. Different doses of ketamine (1–25 mg/kg) were used to generate a dose-response curve of the total distance the animal traveled during a recording session (Supplementary Fig. 3). Angular acceleration (radians/s$^2$) was also calculated to quantify the stereotypic post-ketamine spinning behavior. The mouse underwent only a single injection with a single concentration of the drug on any given day. The number of lifetime injections was limited to 20 per animal and no more than 5 injections per week (one per day).

### Imaging of the hippocampus in the open field arena

Five male and three female Ai94;Camk2a-tTA;Camk2a-Cre (JAX id: 024115 and 005359) mice were used for imaging experiments in region CA1 of the hippocampus. Mice were 8–12 weeks old at the time of surgery. Detailed surgical procedures are described in Sun and Giocomo,[96]. Briefly, after fully anesthetizing each mouse, a gradient refractive index (GRIN) lens (0.25 pitch, 0.55 NA, 1.8 mm diameter and 4.31 mm in length, Edmund Optics) was implanted above the CA1 region of the hippocampus after aspirating the overlying cortical tissue. The implantation coordinates were centered at −2.30 mm anterior/posterior, +1.75 mm medial/lateral and −1.53 mm dorsal/ventral relative to bregma. Two weeks after the implantation of the GRIN lens, a small aluminum base plate was cemented to the animal's head on top of the existing dental cement. After the installation of the baseplate, the imaging window was fixed for long-term use with respect to the miniscope used during installation. For all imaging experiments, each mouse had a dedicated miniscope. When not imaging, a plastic cap was placed in the baseplate to protect the GRIN lens from dust and dirt. After mice had fully recovered from the baseplate surgery, they were handled and allowed to habituate to wearing the head-mounted miniscope by freely exploring an open arena (a clean cage bottom) in a different room for 20 min every day for 1 week. Animals were also habituated to mock IP injections (needle poking) once a day for 4 days.

Before the imaging experiment, mice were further habituated in the circular open field (36 cm in diameter) 20 min/day for 2 days in the experimental room. The imaging experiment then followed an AABA paradigm over 4 days, in which A represents two baseline sessions and a post-baseline session on days 1, 2, and 4, respectively; and B represents the ketamine session on day 3. Each session was 20 min long and mice were allowed to freely explore the circular open field arena. Ketamine (25 mg/kg) was injected IP on day 3 just before imaging began.

### Miniscope imaging data acquisition and initial batch processing

Details of the general processing steps for the customized miniscope acquired data are described in Sun and Giocomo[96]. Briefly, miniscope videos were acquired using custom software (https://github.com/daharoni/Miniscope_DAQ_Software). Individual sessions were first concatenated and down-sampled by a factor of 2 using custom MATLAB scripts, then motion corrected using the NoRMCorre MATLAB package. To align miniscope videos across different sessions for the entire experiment, we applied an automatic 2D image registration method (github.com/fordanic/image-registration) with rigid x-y translations according to the maximum intensity projection images for each session. The registered videos for each animal were then concatenated together in chronological order to generate a combined data set for extracting calcium activity. This process made it unnecessary to perform individual footprint alignment or cell registration across sessions. We then used the CNMF-E package to extract an individual neuron's calcium activity for the concatenated video data. CNMF-E is based on the CNMF framework, which enables simultaneous de-noising, de-convolving and de-mixing of calcium imaging data. These extracted calcium signals for the combined data set were then split back into each session according to their individual frame numbers.

### Place cell analyses

To calculate spatial rate maps, the position and speed of the animal was determined by applying a custom MATLAB script to the animal's behavioral tracking video. Time points at which the speed of the animal was lower than 2 cm/s were identified and excluded from further analysis. We then used linear interpolation to temporally align the position data to the calcium imaging data. After we obtained the deconvolved spiking activity of neurons, we extracted and binarized the effective neuronal calcium events from the deconvolved spiking activity by applying a threshold (3 × standard deviation of all the

deconvolved spiking activity for each neuron). We then treated the binarized deconvolved spikes as calcium events. The position data was sorted into 1.75 × 1.75 cm non-overlapping spatial bins. The spatial rate map for each neuron was constructed by dividing the total number of calcium events by the animal's total occupancy in a given spatial bin. The rate maps were smoothed using a 2D convolution with a Gaussian filter that had a standard deviation of 2.

To identify place cells, and quantify the information content of a given neuron's activity, we calculated spatial information scores in bits/s (each calcium event is treated as a spike here) for each neuron according to the previously published formula[90]. Bins with total occupancy time of less than 0.1 s were excluded from the calculation. To identify place cells, the timing of calcium events for each neuron was circularly shuffled 1000 times and spatial information (bits/sec) recalculated for each shuffle. This generated a distribution of shuffled information scores for each individual neuron. The value at the 95th percentile of each shuffled distribution was used as the threshold for classifying a given neuron as a place cell, and we excluded cells with an overall mean calcium event rate less than 0.1 Hz. This threshold was roughly equal to the 5th percentile of the mean event rate distribution for all neurons.

### Position-matching for comparisons of cell activity across sessions

Analyses that compare hippocampal neuronal activity across different sessions (longitudinal comparisons) can be influenced by biases in the animal's spatial occupancy, which can arise due to the effects of ketamine. To circumvent the effect of differences in occupancy on our analyses, we implemented a position-matched down-sampling protocol when performing longitudinal comparisons of place cell activity. For down-sampling, we first binned the spatial arena into 1.75 × 1.75 cm non-overlapping bins. We then computed the number of position samples (frames) observed in each spatial bin for the to-be-matched sessions. Finally, the number of samples in each corresponding spatial bin were down-sampled by randomly removing position samples, and the corresponding neural activity, from the session with greater occupancy. Due to the stochastic nature of the down-sampling process, we repeated this procedure 50 times (unless otherwise specified) for each cell, and the final value for each cell was calculated as the average of all 50 iterations. This final value was then used to obtain the reported means or perform statistical comparisons. This protocol was applied to our analyses for all the within subject comparisons including mean and peak calcium event rates, spatial information and spatial correlations. To avoid over-downsampling using many sessions simultaneously, we only performed pairwise downsampling with two sessions at a time.

### Reporting summary
Further information on research design is available in the Nature Portfolio Reporting Summary linked to this article.

## Data availability
The MEC binned firing rates, hippocampal calcium traces, and mouse position data generated in this study have been deposited in the Figshare database under accession code https://doi.org/10.6084/m9.figshare.22696309 [https://figshare.com/articles/dataset/Ketamine_evoked_disruption_of_entorhinal_and_hippocampal_spatial_maps/22696309][97]. Source data are provided with this paper.

## Code availability
Code for replicating the analyses in this study have been deposited on GitHub at https://github.com/GiocomoLab/Masuda_et_al_2023 and deposited in the Zenodo database under the accession code: https://doi.org/10.5281/ZENODO.7903214 [https://zenodo.org/record/7903214][98].

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

## Acknowledgements

We thank A. Diaz for histology assistance and animal care, J. Maturi for assistance with open field behavioral training, J. Wen for assistance with virtual reality behavioral training, and Giocomo Lab members for discussions and feedback. This work was supported by the Stanford University Medical Scientist Training Program (T32-GM007365 and T32-GM145402) (F.K.M.), the A.P. Giannini Foundation (E.A.A.J.); NIDA (K01DA058743) (Y.S.); NIH Brain Initiatives (U19NS118284), NIMH (MH126904 and MH130452), the Simons Foundation (SCGB 542987SPI), NIDA (DA042012), the Vallee Foundation, and the James S. McDonnell Foundation (L.M.G.).

## Author contributions

F.K.M.: conceptualization, methodology, formal analysis, investigation, writing—original draft, writing—review and editing, and visualization; Y.S.: methodology, formal analysis, investigation, writing—original draft, writing—review and editing, and visualization; E.A.A.J.: formal analysis, investigation, writing—original draft, writing—review and editing, and visualization; L.M.G.: conceptualization, methodology, writing—original draft, writing—review and editing, visualization, supervision, funding.

## Competing interests

The authors declare no competing interests.
