## [Peer Review File · Nature Communications]

Ketamine evoked disruption of entorhinal and hippocampal spatial mapsREVIEWER COMMENTS

Reviewer #1 (Remarks to the Author):

Masuda and colleagues report on the effects of ketamine on the spatial coding characteristics of hippocampal and entorhinal cortical neurons.

They present profound effects on the spatial coding properties of these neurons.

The work is elegant and the figures nicely demonstrate the observed phenomenon.

Below however, I outline some needed revisions, largely to the presentation of the introduction and discussion that would allow for a story to be presented that integrates with the existing literature on ketamine's effect on hippocampal physiology and well-known disruption of memory and cognition, including spatial cognition that would highlight the particular contribution/insights of the presented research.

First/foremost, the fact that the researchers use a technically powerful set of tools to demonstrate that ketamine, disrupts the spatial coding characteristic of HPC/EC neurons should NOT be the main message of this manuscript. There has to be a more enlightening message.

1A) Specifically, the abstract can't end on "these findings point to disruption in the spatial coding properties of the entorhinal-hippocampal circuit as a potential neural substrate for ketamine-induced changes in spatial cognition". This statement suggests the research exists in an ahistorical vacuum of research examining ketamine on hippocampal physiology and the spatial memory/cognitive deficits readily observed following ketamine treatment in rodents/humans.

1B) Introduction-2nd paragraph. This paragraph could be reduced in length. It takes up a third of the introduction, and rehearses in-depth well-known material about spatial coding, without much enlightening about ketamine and its well-described effects on HPC physiology and memory function.

1C) Introduction. This last paragraph mostly highlights that the researchers used an impressive array of tools to demonstrate what might generally have been expected with a less impressive array of tools.

This is not to say it should not be highlighted, but less grandstanding, and more attention to what the new tools might/could reveal would be appreciated.

1D) Discussion-1st paragraph. Starting with “The technical challenge.....and the hippocampus in navigating animals. I would delete those sentences or revise and move later in the discussion. Preferably delete as they once again highlight that the paper reports using an impressive array of tools, to demonstrate what might be expected given knowledge of how ketamine alters HPC physiology in behaving rodents (see references below). Better to highlight what was revealed by using this impressive array of tools than high-lighting the importance of the tools.

1E) Discussion-last paragraph. In my opinion, the paragraph does NOT do justice to the research and doesn't highlight the value of the research, rather rests on platitudes about how “future work” and the “advent of new technologies” will solve problems (which they may) but what does the current research say? I would rework/start over on this entire paragraph. While it is true that other associative regions (retrosplenial) support spatial cognition as do many other circuits conveying input/communicating to HPC/EC, the rationale for highlighting here and integrating it into a less than clear-messaged paragraph is not compelling.

Second, the authors fail to reference or place into context a number of potential lines of research that examine the effects of ketamine on spatial memory, cognition in general, and studies examining the effects of ketamine on hippocampal physiology.

2A) The authors highlight the role of ketamine on HCN1, more so than the effects of ketamine on NMDA, which may or not be understandable, but if they are going to highlight the HCN1 effects, then they should link to manipulation of theta/gamma. The authors may want to point to how the primary author's (Giocomo et al., 2007 and Giocomo and Hasselmo, 2009) prior research links to that of Alonso and Llinas (1989) and alteration of theta/gamma rhythms in HPC/EC (see Quilichini, Sirota and Buzsaki, 2010 for inspiration).

2B) In keeping with the above there are several papers that evidence ketamine alterations of hippocampal theta, gamma and cross-frequency coupling at similar or LOWER doses of ketamine than used in the current paper (see additional note below). These include Lazarewicz et al., 2009, Caixeta et al, Sci Rep 2013; Hinman et al., 2013 among others. There are also any number of reviews that also overview much of this research for example, Ahnaou and colleagues (2017) in Translation Psychiatry and Speers and Bilkey (2021) Frontiers Neural Circuits.

3B) It is not clear why the only mention of theta is in reference to Skaggs and colleagues (1996) which was the first clear presentation, that spatial coding was directly related to theta phase. Given that

ketamine disrupts theta and theta/gamma coupling at doses much lower than reported in the current manuscript, there should be some acknowledgement of this well-known feature of hippocampal physiology (see Michaels and colleagues 2018 EJN for review and references).

On that note, I don't think gamma is mentioned in the manuscript.

4B) It is generally a truism that sensitive behavioral/cognitive indices are much more sensitive to subtler manipulations of CNS function such as dose-related effects on indices compared to neurochemical, molecular, or physiological. Thus, it is not surprising that quality behavioral/cognitive indices (as evidenced in the supplemental figures of the present manuscript) are much more sensitive to lower doses of ketamine [(0.5--25 mg/kg in humans/rodents) see also for references Gill et al., *Neurosci Biobehav Rev* 2021; reviewed in Morgan and Curran (2006 as cited in present manuscript) and also Chrobak et al., 2008 for spatial memory deficits in rodents at 2.5mg/kg].

My point would be that somewhere in the neurophysiological data and advantage purported provided by recording hundreds of MEC and HPC there might be indices more sensitive than effects at 25mg and 50mg of ketamine. If there is a technological/analytic advantage in the current approach then further analysis of large data sets or some of those data sets in relation to theta/gamma indices should provide more power to detect relevant change at doses closer to those behaviorally/cognitive relevant to (0.5-10 mg/kg). Otherwise, it is not clear the advantage.

Last, while I don't expect the authors to incorporate all the possible threads/references noted above, I think they can do a much better job of highlighting what they observed and how it enhances the field beyond, we recorded a lot more neurons at once and that is somehow more advantageous. Some of the references noted in this review may, or/not, provide insight in this regard. Importantly, I think there has to be more of a story presented within the context of previous literature and I would suggest looking for relations of spatial coding to theta/gamma indices would be a good place to start or minimally acknowledging that it may. If the authors think that the relationship of coding to theta/gamma would not be revealing then that should perhaps be explicated.

Reviewer #2 (Remarks to the Author):

This paper is a fascinating study about the ketamine-induced disruption of the neural underpinnings of entorhinal-hippocampal spatial maps.

One of the most interesting aspects of this study is the possibility that the phenomena described help to explain dissociative symptoms produced by ketamine. Clearly, ketamine produces dose-related alterations in body perception (altered size and movement of body parts), the environment (distortion of sizes, color, sound, and movement [walls breathing, etc.]), slowed sense of time, and altered perception of one's body in relation to the spatial environment. Perhaps at a "meta level" people feel that things are "unreal" and that their personhood is altered (depersonalization, some feel that body parts have been replaced with machine parts or that others around them have been replaced by robots or doubles). Having observed well over 1000 ketamine infusions in the traditional subanesthetic dosing, I have not observed someone have a clear out-of-body experience as described. I have wondered whether these reports come from people who are nearly anesthetized, perhaps asleep. At 0.2 mg/kg, i.v., over 40 min. (perhaps 90 ng/ml) ketamine does not produce dissociation. Antidepressant effects are present around 0.5 mg/kg (150-200 ng/ml) and patients show increasing reductions in responsiveness above 1.0 mg/kg (>300 ng/ml) as you get to anesthesia. Because ketamine's effects on dissociation have such exquisite dose sensitivity in the subanesthetic range, any help in linking the doses studied here to human doses would be helpful.

The increase in MEC excitatory changes without commensurate reductions in local GABA neuronal activity might suggest that the pyramidal neurons are being driven by extrinsic excitatory input. However, the data seem to point to a loss of inhibitory tuning (excitatory neurons firing across the track). How are these findings reconciled? Similarly, reductions in cell-cell connectivity could reflect an inhibitory effect of E-E connectivity in feedforward networks, as you seen in PFC (Wang et al. *Neuron* 2013) or it could reflect a loss of inhibitory tuning via disrupting E-I connectivity (Murray et al. *Cerebral Cortex* 2014), or both. It would be helpful to understand the drivers of reductions in functional connectivity of the neuron pairs.

The effects of ketamine on E and I cells in hippocampus were not clear to me in this study. Prior studies of NMDAR antagonist effects in hippocampus (Grunze et al. *J Neurosci* 1996; Maccaferri and Dingledine *J Neurosci* 2002; Izumi and Zorumski *Neuropharm* 2014, others) suggested inhibition/modulation of interneuron subpopulations and pyramidal neuronal disinhibition.

The discussion also raises the issue of dose-related ketamine effects. Ketamine is most potent at NR2C/2D-containing NMDAR, about half as potent at NR2B/2A-containing NMDARs, and somewhere between 1/5-1/10 as potent at HCN1. It is interesting that the discussion suggests that ketamine effects in MEC are not present in HCN1 KO mice. However, in reference 54 (Chen et al. *J Neurosci* 2009), R,S-

ketamine produced 50% inhibition of HCN1 at 100 μ M, i.e., well into the anesthetic range and the behavioral response measured was loss-of-righting, i.e., anesthesia. This suggests that any alteration in subanesthetic ketamine response in HCN1 KO mice might be due to synaptic adaptations to developmental loss of HCN1 rather than a direct consequence of loss of HCN1.

Reviewer #3 (Remarks to the Author):

The manuscript by Masuda et al addressed the effects of sub-anesthetic ketamine on the firing of medial entorhinal cortex (MEC) and hippocampus (H) neurons during navigation. They show that ketamine disrupts behavior on a simple (linear track) VR task and on object location memory task. Consistent with the well established role of MEC and H circuits in spatial navigation, they show that ketamine disrupts firing patterns of MEC and H neurons. Specifically, they demonstrate that immediately upon injection of ketamine MEC activity de-coheres and subsequently, as ketamine concentration wanes, the MEC activity undergoes remapping. They complemented their work with neuropixels probes in MEC with miniscope recordings of calcium transients in H. There they find a slightly different pattern of neuronal perturbations caused by ketamine. Specifically, they show an overall suppression of neuronal activity, and a decrease in the number of cells that were classified as place cells.

Overall, given the behavioral effects of ketamine on navigation behavior and the known role of MEC and H circuits in navigation, the fact that ketamine alters activity within these circuits is not surprising. Nevertheless, I do think that the state-of-the-art methods used by the authors to characterize how ketamine alters population activity are both novel and interesting. This is why I think that the paper should be published after some minor revisions.

1. Ketamine, at sub-anesthetic doses produces visual disturbances (e.g. Oye, I., Ole Paulsen, and Atle Maurset. "Effects of ketamine on sensory perception: evidence for a role of N-methyl-D-aspartate receptors." *Journal of Pharmacology and Experimental Therapeutics* 260.3 (1992): 1209-1213., Vollenweider, F. X., et al. "Differential psychopathology and patterns of cerebral glucose utilisation produced by (S)- and (R)-ketamine in healthy volunteers using positron emission tomography (PET)." *European Neuropsychopharmacology* 7.1 (1997): 25-38.). The authors should discuss the possibility that the differences in MEC activity patterns are not necessarily due directly to the disruptions of the spatial navigation circuitry per se but due to the corrupted visual inputs. In proper (not VR) navigation, behaviors are likely driven by combination of visual and proprioceptive inputs. But proprioceptive hallucinations are also highly common under sub-anesthetic ketamine.

2. I found the characterization of the grid cells a bit lacking. It seems that the authors identified grid cells as those neurons whose firing rate was modulated by varying the coupling between mouse motion and the optical flow. I understand that this method could be used to identify some putative grid cells, but as the authors acknowledge, this is not perfect. Why not use some standard "gridness" score? There is a missed opportunity here to characterize not only behavior of individual grid cells (how periodic they are

in space etc) but also the spatial scaling of different grid cells. This could offer some insights into how ketamine disrupts the MEC circuit.

3. On a related note, I was not certain why the authors chose to implement a 1D track for their virtual reality. I appreciate the simplicity of the behavior in this setting, but for the purposes of mapping grid and place cells, one does not necessarily need a behavioral task. The authors should comment on this.

4. This is a minor point, but I was not sure why the authors used both PCA and UMAP to cluster neuronal activity. It seems that UMAP is a bit better which is not terribly surprising. So, why do they also need PCA results?

We thank the reviewers for their helpful comments and constructive criticism. We have made changes to the text, analyses and figures in accordance with their suggestions. We feel that these changes strengthen the paper's conclusions and broaden its impact. In response to reviewer comments, we have added new text to the introduction, discussion (three new paragraphs), and made several text clarifications. We have also added three additional Supplementary Figures (Supplementary Figs. 4, 6, and 7), and ~40 additional references.

We have also provided source data and in the process, added non-parametric statistical tests and exact p-values to many of the comparisons, which we have noted where applicable in the manuscript and legends. This process also resulted in a few very minor corrections. Below, we provide a detailed response to each Reviewer suggestion. All major changes made in the revised article are highlighted in **gray** in the manuscript file.

Reviewer #1:

Masuda and colleagues report on the effects of ketamine on the spatial coding characteristics of hippocampal and entorhinal cortical neurons. They present profound effects on the spatial coding properties of these neurons. The work is elegant and the figures nicely demonstrate the observed phenomenon. Below however, I outline some needed revisions, largely to the presentation of the introduction and discussion that would allow for a story to be presented that integrates with the existing literature on ketamine's effect on hippocampal physiology and well-known disruption of memory and cognition, including spatial cognition that would highlight the particular contribution/insights of the presented research.

We thank the reviewer for their constructive feedback and agree that the work could be better framed in the context of what has been previously shown regarding the effects of ketamine on hippocampal physiology, memory and cognition. To address this concern, we have re-written portions of the Abstract and added new text to both the Introduction and Discussion, which we believe re-frames the work to better reflect the broader context of the literature. New text is highlighted in gray in the revised manuscript.

First/foremost, the fact that the researchers use a technically powerful set of tools to demonstrate that ketamine disrupts the spatial coding characteristic of HPC/EC neurons should NOT be the main message of this manuscript. There has to be a more enlightening message. 1A) Specifically, the abstract can't end on "these findings point to disruption in the spatial coding properties of the entorhinal-hippocampal circuit as a potential neural substrate for ketamine-induced changes in spatial cognition". This statement suggests the research exists in an ahistorical vacuum of research examining ketamine on hippocampal physiology and the spatial memory/cognitive deficits readily observed following ketamine treatment in rodents/humans.

We agree with the reviewer that the prior version of the manuscript was overly focused on the technology. We have removed the sentence from the abstract described in this comment and have amended the abstract in a way we hope better frames the take home messages of the work. In addition, we have removed text from the introduction, added new text to the introduction and added significantly more text to the discussion – all of which we hope better encompasses the prior work on ketamine in these circuits, provides context for some of the new analyses we have included (e.g. ketamine's impact on entorhinal theta/gamma oscillations) and better frames the current work in the larger picture of what is known/unknown about ketamine's impact on these circuits.

1B) Introduction-2nd paragraph. This paragraph could be reduced in length. It takes up a third of the introduction, and rehearses in-depth well-known material about spatial coding, without much enlightening about ketamine and its well-described effects on HPC physiology and memory function.

We have reduced the text in this paragraph and have added an Introduction paragraph summarizing the previous literature on ketamine's effect on hippocampal physiology and spatial memory.

1C) Introduction. This last paragraph mostly highlights that the researchers used an impressive array of tools to demonstrate what might generally have been expected with a less impressive array of tools. This is not to say it should not be highlighted, but less grandstanding, and more attention to what the new tools might/could reveal would be appreciated.

We have removed the text that was focused on the tools and instead highlighted what was revealed by the tools - ketamine-induced alterations in spatial memory at the level of behavior, single cells, and neural populations in two interconnected regions critical to spatial navigation.

1D) Discussion-1st paragraph. Starting with "The technical challenge.....and the hippocampus in navigating animals. I would delete those sentences or revise and move later in the discussion. Preferably delete as they once again highlight that the paper reports using an impressive array of tools, to demonstrate what might be expected given knowledge of how ketamine alters HPC physiology in behaving rodents (see references below). Better to highlight what was revealed by using this impressive array of tools than high-lighting the importance of the tools.

We have made similar changes to the text of the Discussion as we have to the Introduction (see point 1C), and this text has been deleted.

1E) Discussion-last paragraph. In my opinion, the paragraph does NOT do justice to the research and doesn't highlight the value of the research, rather rests on platitudes about how "future work" and the "advent of new technologies" will solve problems (which they may) but what does the current research say? I would rework/start over on this entire paragraph. While it is true that other associative regions (retrosplenial) support spatial cognition as do many other circuits conveying input/communicating to HPC/EC, the rationale for highlighting here and integrating it into a less than clear-messaged paragraph is not compelling.

We have expanded the Discussion, adding three paragraphs along with several other text additions, to better highlight the value of the research. In this paragraph, our goal was to highlight that because several brain regions are affected by ketamine in ways that would disrupt spatial navigation, we cannot disentangle the roles of each until future work locally infuses ketamine into specific regions. To better highlight our own results as the reviewer suggests, we have reduced the text about the retrosplenial cortex. We have also added a sentence highlighting how future work would answer specific questions raised by the current research.

Second, the authors fail to reference or place into context a number of potential lines of research that examine the effects of ketamine on spatial memory, cognition in general, and studies examining the effects of ketamine on hippocampal physiology.

We have added new references, text to the Introduction and Discussion, as noted above.

2A) The authors highlight the role of ketamine on HCN1, more so than the effects of ketamine on NMDA, which may or not be understandable, but if they are going to highlight the HCN1 effects, then they should

link to manipulation of theta/gamma. The authors may want to point to how the primary author's (Giocomo et al., 2007 and Giocomo and Hasselmo, 2009) prior research links to that of Alonso and Llinas (1989) and alteration of theta/gamma rhythms in HPC/EC (see Quilichini, Sirota and Buzsaki, 2010 for inspiration).

We have added a sentence to the Discussion paragraph about HCN1 channels to link it to the theta and gamma changes observed in the literature, as noted by the reviewer, and that we observed in the new analyses. We decided to keep this discussion focused on extracellularly measured theta oscillations, as the link between membrane oscillations (Giocomo et al., 2007; Giocomo & Hasselmo, 2009; Alonso and Llinas, 1989) is still not entirely resolved (e.g. Fernandez & White, J Neurosci, 2008 show that high conductance states like those in vivo reduce subthreshold membrane oscillation dynamics substantially). However, if the reviewer or editor feels including further discussion on the interactions of membrane oscillations, HCN1 channels, extracellularly measured theta oscillations and ketamine, we would be happy to do so via additional text revisions.

2B) In keeping with the above there are several papers that evidence ketamine alterations of hippocampal theta, gamma and cross-frequency coupling at similar or LOWER doses of ketamine than used in the current paper (see additional note below). These include Lazarewicz et al., 2009, Caixeta et al, Sci Rep 2013; Hinman et al., 2013 among others. There are also any number of reviews that also overview much of this research for example, Ahnaou and colleagues (2017) in Translation Psychiatry and Speers and Bilkey (2021) Frontiers Neural Circuits.

We agree with the reviewer that better consideration of how ketamine influences theta and gamma is needed. First, we added new text to the Introduction and Discussion to better consider our findings in the context of prior work on ketamine and oscillatory activity. Second, we performed a set of new analyses to verify that we observe the same effects on theta and gamma oscillations in MEC as those reported in prior works. These new analyses are now presented in Supplementary Figure 4 and agree with observations in the hippocampus of a reduction in theta/increase in gamma following ketamine administration (Lazarewicz et al., 2009, Caixeta et al, Sci Rep 2013; Hinman et al., 2013). We have also added text describing these analyses and findings to the Results section.

3B) It is not clear why the only mention of theta is in reference to Skaggs and colleagues (1996) which was the first clear presentation, that spatial coding was directly related to theta phase. Given that ketamine disrupts theta and theta/gamma coupling at doses much lower than reported in the current manuscript, there should be some acknowledgement of this well-known feature of hippocampal physiology (see Michaels and colleagues 2018 EJM for review and references). On that note, I don't think gamma is mentioned in the manuscript.

We have added text to the Discussion text regarding how our observed decrease in theta power might affect spatial coding. We have also added an analysis of fast gamma power (see response to point 2C).

4B) It is generally a truism that sensitive behavioral/cognitive indices are much more sensitive to subtler manipulations of CNS function such as dose-related effects on indices compared to neurochemical, molecular, or physiological. Thus, it is not surprising that quality behavioral/cognitive indices (as evidenced in the supplemental figures of the present manuscript) are much more sensitive to lower doses of ketamine [(0.5--25 mg/kg in humans/rodents) see also for references Gill et al., Neurosci Biobehav Rev 2021; reviewed in Morgan and Curran (2006 as cited in present manuscript) and also Chrobak et al., 2008 for spatial memory deficits in rodents at 2.5mg/kg]. My point would be that somewhere in the neurophysiological data and advantage purported provided by recording hundreds of MEC and HPC there might be indices more sensitive than effects at 25mg and 50mg of ketamine. If there is a technological/analytic advantage in the current approach then further analysis of large data sets or

some of those data sets in relation to theta/gamma indices should provide more power to detect relevant change at doses closer to those behaviorally/cognitive relevant to (0.5-10 mg/kg). Otherwise, it is not clear the advantage.

We agree with the reviewer that more data does not necessarily provide a clear advantage. Here, we used high density recordings primarily to: (1) record many neurons in an individual session and individual animal. As pharmacological manipulations can result in variability across animals and sessions, large numbers of neurons recorded in each session provides a powerful tool to establish reproducibility. We also used high density recordings to (2) perform analyses that require many cells per session and (3) investigate how ketamine impacts sub-populations of functionally defined neurons (such as grid cells), which are thought to play a key role in spatial processes but comprise a small proportion of the total MEC cell population and are thus challenging to record in large numbers using technological approaches such as tetrodes. To better emphasize these strengths, we: (1) note the consistency in the effect of ketamine in individual animals in the Results section, where applicable, (2) expanded our cell-pair analyses to include transmission probability (new Supplementary Figure 6), which was possible because we were able to identify many putative monosynaptic connections due to the number of cells we were able to record from simultaneously, and (3) expanded our analyses on grid cells (new Supplementary Figure 7), which were only possible because we were able to identify 108 strictly defined grid cells out of 2894 excitatory neurons.

We would also note that we chose 25 mg/kg as we felt it was a behaviorally and cognitively relevant dose. Many ketamine behavior and neural experiments use this dose or greater. Intraperitoneal injection of 150 mg/kg (Pal et al, British Journal of Anesthesia, 2015) or 200 mg/kg (Ganguly et al, Drug Metabolism and Disposition, 2018) ketamine have been shown to induce anesthesia (loss of righting reflex) in mice, while lower doses do not. Sensory dissociation (e.g. measured through reduced response to noxious stimuli) in mice is observed at intraperitoneal doses 25-100 mg/kg, with no dissociation at doses 13 mg/kg and below (Cichon et al, Nature Neuroscience, 2023 and Vesuna et al, Nature, 2020). This suggests the anesthetic dose range is ~150+ mg/kg and the subanesthetic dissociative range is ~25-100 mg/kg. Our dose was chosen to fall at the lower end of the dissociative range. We have added a sentence to the first paragraph of the Results to provide this context and added text to the Discussion to add context of the different effects of ketamine at different doses.

Last, while I don't expect the authors to incorporate all the possible threads/references noted above, I think they can do a much better job of highlighting what they observed and how it enhances the field beyond, we recorded a lot more neurons at once and that is somehow more advantageous. Some of the references noted in this review may, or/not, provide insight in this regard. Importantly, I think there has to be more of a story presented within the context of previous literature and I would suggest looking for relations of spatial coding to theta/gamma indices would be a good place to start or minimally acknowledging that it may. If the authors think that the relationship of coding to theta/gamma would not be revealing then that should perhaps be explicated.

We agree with the reviewer that the manuscript needed to better incorporate prior work on ketamine and its effects on oscillatory dynamics, spatial coding and cognition. As noted in our prior responses, we have added a new Supplementary Figure in response to point 2B and now examine theta and gamma band oscillations in MEC after ketamine administration. We have also added physiological contexts in terms of what prior papers have reported on the effects of ketamine on theta and gamma to the Introduction, Results and Discussion sections of the manuscript. In particular, we added to paragraphs to the Discussion: one about how theta affects spatial coding and how fast gamma may contribute to synchronization between regions, and one about why we observed an increase in MEC excitatory cell activity but a decrease in CA1 excitatory cell activity.

Reviewer #2:

This paper is a fascinating study about the ketamine-induced disruption of the neural underpinnings of entorhinal-hippocampal spatial maps.

We thank the reviewer for their comments and complimentary overall description of the work.

One of the most interesting aspects of this study is the possibility that the phenomena described help to explain dissociative symptoms produced by ketamine. Clearly, ketamine produces dose-related alterations in body perception (altered size and movement of body parts), the environment (distortion of sizes, color, sound, and movement [walls breathing, etc.]), slowed sense of time, and altered perception of one's body in relation to the spatial environment. Perhaps at a "meta level" people feel that things are "unreal" and that their personhood is altered (depersonalization, some feel that body parts have been replaced with machine parts or that others around them have been replaced by robots or doubles). Having observed well over 1000 ketamine infusions in the traditional subanesthetic dosing, I have not observed someone have a clear out-of-body experience as described. I have wondered whether these reports come from people who are nearly anesthetized, perhaps asleep. At 0.2 mg/kg, i.v., over 40 min. (perhaps 90 ng/ml) ketamine does not produce dissociation. Antidepressant effects are present around 0.5 mg/kg (150-200 ng/ml) and patients show increasing reductions in responsiveness above 1.0 mg/kg (>300 ng/ml) as you get to anesthesia. Because ketamine's effects on dissociation have such exquisite dose sensitivity in the subanesthetic range, any help in linking the doses studied here to human doses would be helpful.

The reviewer raises an important and highly interesting point. In particular, given the recent increase in the use of ketamine in clinical settings, understanding how effects in rodents may correspond to humans is important for informing future studies.

First, we have changed the text in the Introduction to better match experiences reported after a human antidepressant dose (0.5 mg/kg/40 minutes), as described in the Clinician Administered Dissociative State Scale (e.g. disconnected from your own body or watching the situation as an observer) (van Schalkwyk et al, *J Affective Disorders*, 2018).

Second, we acknowledge the challenge of interspecies scaling. Even so, if we assume ~50% bioavailability of ketamine via a 25 mg/kg IP dose in mice, this should correspond to ~1 mg/kg i.v. in a human, which would capture the reports of changes in spatial cognition after a human antidepressant dose of 0.5 mg/kg (From Guidance for Industry, Estimating the maximum safe starting dose in initial clinical trials for therapeutics in adult health volunteers, Food and Drug Administration, July 2005).

Finally, if we consider our dosage from the behavioral perspective, prior work has shown that mice produce behaviors consistent with sensory dissociation (reduced response to noxious stimuli) at intraperitoneal doses 25-100 mg/kg, with no dissociation at doses 13 mg/kg and below (Cichon et al, *Nature Neuroscience*, 2023 and Vesuna et al, *Nature*, 2020). On the other hand, intraperitoneal injection of 150 mg/kg (Pal et al, *British Journal of Anesthesia*, 2015) or 200 mg/kg (Ganguly et al, *Drug Metabolism and Disposition*, 2018) ketamine have been shown to induce anesthesia (loss of righting reflex) in mice, while lower doses do not. This suggests the anesthetic dose range is 150+ mg/kg and the subanesthetic dissociative range is 25-100 mg/kg. We chose our dose to lie at the lower end of the dissociative range so that we could maximally preserve the locomotor behavior needed to examine spatial coding while also considering ketamine at a dosage associated with dissociation. We have added a sentence to the first paragraph of the Results to provide this context and added text to the Discussion to provide better context regarding the different effects of ketamine at different doses.

The increase in MEC excitatory changes without commensurate reductions in local GABA neuronal activity might suggest that the pyramidal neurons are being driven by extrinsic excitatory input. However, the data seem to point to a loss of inhibitory tuning (excitatory neurons firing across the track). How are these findings reconciled? Similarly, reductions in cell-cell connectivity could reflect an inhibitory effect of E-E connectivity in feedforward networks, as you seen in PFC (Wang et al. Neuron 2013) or it could reflect a loss of inhibitory tuning via disrupting E-I connectivity (Murray et al. Cerebral Cortex 2014), or both. It would be helpful to understand the drivers of reductions in functional connectivity of the neuron pairs.

The reviewer raises an interesting point. Based on this comment, we decided to add a transmission probability analysis (Supplementary Fig. 6) to dissect whether this increase in excitatory neuron activity was due to disinhibition from local inhibitory neurons or increased intrinsic excitability of excitatory neurons. We identified putative monosynaptically connected pairs of neurons based on their cross-correlograms (using methods from Fernandez-Ruiz et al, Neuron, 2017), then classified them as excitatory or inhibitory (Supplementary Fig. 6a-b). We calculated the extent to which a single spike from the upstream cell in each pair (cell A) changed the probability of spiking from the downstream cell (cell B) in the 5 ms following the spike, compared to the 5 ms prior to the spike. This transmission probability increased in excitatory neurons and decreased in inhibitory neurons following ketamine injection, thus both excitatory and inhibitory cells were more effective at altering the activity of their downstream partners (Supplementary Fig. 6c-e). This suggests that the increase in excitatory neuron firing was not due to disinhibition (e.g. via antagonizing NMDAR on GABAergic neurons), but at least in part due to increased effectiveness of excitatory synaptic transmission, perhaps in combination with increased intrinsic excitability and increased external inputs. We did not observe any difference dependent on the putative of the downstream neuron (i.e. E-E and E-I transmission probabilities both increased and I-E and I-I both decreased). Interestingly, while the probability of transmission across a monosynaptic pair on the timescale of 5 ms increases following ketamine administration, the coordination between spikes times on the timescale of a single trial (Fig. 5) declines. This points to a loss of coordination of spike timing, which could be due to the loss of theta oscillatory power we observed in our recent addition to the manuscript (Supplementary Fig. 4).

The effects of ketamine on E and I cells in hippocampus were not clear to me in this study. Prior studies of NMDAR antagonist effects in hippocampus (Grunze et al. J Neurosci 1996; Maccaferri and Dingledine J Neurosci 2002; Izumi and Zorumski Neuropharm 2014, others) suggested inhibition/modulation of interneuron subpopulations and pyramidal neuronal disinhibition.

All hippocampal results were in excitatory cells in CA1, as the calcium indicator was driven by the CAMKII promoter. The mean and peak calcium event rate declined following ketamine administration, suggesting pyramidal cell inhibition. Several studies (Wakasugi et al Anesth Analg 1999, Gage et al Br J Pharm 1985, Aida et al Br J Pharm 1994, Jang & MacIver Int J Mol Sci 2021) have shown that *ex vivo* application of ketamine at concentrations ranging from 1 μ M to 500 μ M on hippocampal slices reduces population spike amplitudes, inhibits spontaneous activity, and prolongs IPSCs. However, one conflicting study has shown that 1 μ M ketamine reduces IPSC frequency, increases AP frequency, and increases bursting in hippocampal slices (Wolman & McMahon PNAS 2018). To our knowledge, this manuscript is the first examination of CA1 pyramidal cell firing following ketamine administration *in vivo*. Our results suggest that CA1 pyramidal cells are inhibited, rather than disinhibited, during acute 25 mg/kg ketamine administration. This may be through its interaction with channels besides NMDAR, such as HCN1. We have added a paragraph to the Discussion to put our results into context of these previous findings, and to interpret them given the opposite effect (increased excitability) we found in MEC.

The discussion also raises the issue of dose-related ketamine effects. Ketamine is most potent at NR2C/2D-containing NMDAR, about half as potent at NR2B/2A-containing NMDARs, and somewhere between 1/5-1/10 as potent at HCN1. It is interesting that the discussion suggests that ketamine effects in MEC are not present in HCN1 KO mice. However, in reference 54 (Chen et al. J Neurosci 2009), R,S-ketamine produced 50% inhibition of HCN1 at 100 μ M, i.e., well into the anesthetic range and the behavioral response measured was loss-of-righting, i.e., anesthesia. This suggests that any alteration in subanesthetic ketamine response in HCN1 KOs might be due to synaptic adaptations to developmental loss of HCN1 rather a direct consequence of loss of HCN1.

The reviewer raises an interesting line of considerations. In the Discussion, we include the (Chen et al. J Neurosci 2009) reference to illustrate that HCN1-KO removes the effect of ketamine on HCN channels, namely inhibiting Ih and thus reducing the resting membrane potential and increasing EPSP summation. The dose used in the Chen et al paper to achieve this result was 20 μ M, the low end of the anesthetic range in rodents. If knocking out HCN1 channels removes this effect of ketamine at 20 μ M, it stands to reason that it would also remove this effect at lower, subanesthetic doses of ketamine. However, we cannot tell from this study if the removed effect in HCN1-KO mice was because of loss of HCN1 channels or synaptic alterations to developmental loss of HCN1, as the reviewer suggests. In a related study (Zhang et al, Sci. Signal, 2016), pharmacological blockade of HCN channels was used to achieve a similar effect in CA1 (removing the EPSC amplitude increase effect of ketamine at 20 μ M), suggesting that HCN1 loss or block removes the effect of ketamine antagonism of HCN1 channels. We had added this reference to this Discussion paragraph for clarity.

The reviewer raises an important point about dose effects however, given the difference in affinity of ketamine for HCN1 vs NMDAR. We've added sentences to the Discussion highlighting that ketamine's effects on HCN1 are mainly tested at anesthetic doses, thus more research at subanesthetic doses is needed.

Reviewer #3:

The manuscript by Masuda et al addressed the effects of sub-anesthetic ketamine on the firing of medial entorhinal cortex (MEC) and hippocampus (H) neurons during navigation. They show that ketamine disrupts behavior on a simple (linear track) VR task and on object location memory task. Consistent with the well established role of MEC and H circuits in spatial navigation, they show that ketamine disrupts firing patterns of MEC and H neurons. Specifically, they demonstrate that immediately upon injection of ketamine MEC activity de-coheres and subsequently, as ketamine concentration wanes, the MEC activity undergoes remapping. They complemented their work with neuropixels probes in MEC with miniscope recordings of calcium transients in H. There they find a slightly different pattern of neuronal perturbations caused by ketamine. Specifically, they show an overall suppression of neuronal activity, and a decrease in the number of cells that were classified as place cells. Overall, given the behavioral effects of ketamine on navigation behavior and the known role of MEC and H circuits in navigation, the fact that ketamine alters activity within these circuits is not surprising. Nevertheless, I do think that the state-of-the-art methods used by the authors to characterize how ketamine alters population activity are both novel and interesting. This is why I think that the paper should be published after some minor revisions.

We sincerely thank the reviewer for their interest in our manuscript and reported findings.

1. Ketamine, at sub-anesthetic doses produces visual disturbances (e.g. Oye, I., Ole Paulsen, and Atle Maurset. "Effects of ketamine on sensory perception: evidence for a role of N-methyl-D-aspartate receptors." *Journal of Pharmacology and Experimental Therapeutics* 260.3 (1992): 1209-1213., Vollenweider, F. X., et al. "Differential psychopathology and patterns of cerebral glucose utilisation produced by (S)-and (R)-ketamine in healthy volunteers using positron emission tomography (PET)." *European Neuropsychopharmacology* 7.1 (1997): 25-38.). The authors should discuss the possibility that the differences in MEC activity patterns are not necessarily due directly to the disruptions of the spatial navigation circuitry per se but due to the corrupted visual inputs. In proper (not VR) navigation, behaviors are likely driven by a combination of visual and proprioceptive inputs. But proprioceptive hallucinations are also highly common under sub-anesthetic ketamine.

The reviewer raises an important point that we had not yet addressed in the Discussion. We have added sentences to the final Discussion paragraph and the references mentioned here to highlight the importance of visual and somatosensory cortex in affecting spatial navigation, and how these hallucinations could affect spatial representations.

2. I found the characterization of the grid cells a bit lacking. It seems that the authors identified grid cells as those neurons whose firing rate was modulated by varying the coupling between mouse motion and the optical flow. I understand that this method could be used to identify some putative grid cells, but as the authors acknowledge, this is not perfect. Why not use some standard "gridness" score? There is a missed opportunity here to characterize not only behavior of individual grid cells (how periodic they are in space etc) but also the spatial scaling of different grid cells. This could offer some insights into how ketamine disrupts the MEC circuit.

First, we have added additional analyses of grid cells to characterize how the MEC circuit is disrupted, as suggested (new Supplementary Fig. 7). As in the larger excitatory neuron population, the mean firing rate increased by ~2.5 Hz in putative grid cells after the ketamine injection, while the control injection had no effect on grid cell firing rates (Supplementary Fig. 7a). As in the larger spatial cell population, grid cell spatial selectivity degraded following ketamine injection (Supplementary Fig. 7b-e). However, unlike in the larger spatial cell

population, grid cells do not recover spatial stability during the ketamine epoch (Supplementary Fig. 7d). We have also added analyses of grid cell field width, which can be measured in 1D virtual reality. In grid cells, we identified peaks in the raster plot averaged over blocks of trials and took the largest peak width as the grid field width (Supplementary Fig. 4f). We found that grid field width did not change following ketamine administration (Supplementary Fig. 4g), suggesting that the reduced spatial selectivity was not due to an increase in grid scale (which directly scales with grid field width).

Second, unfortunately a standard 'gridness' score is not possible in a 1D environment. The classic 'gridness score' relies on metrics derived from an autocorrelation of the 2D grid cell firing pattern (first used in Langston et al., Science, 2010). In head-fixed virtual reality, grid cell firing patterns have not been observed in 2D environments, likely due to the mismatch between vestibular and visual movements (personal communications from Edvard Moser, Daniel Dombeck and Dora Angelaki). In freely moving 2D environments, spatial cells are nearly impossible to observe, as ketamine induces a stereotyped spinning behavior (Supplementary Fig. 3e) that degrades the firing rate maps to the point that metrics like a 'gridness score' cannot be calculated. Even so, in our prior work we demonstrated that grid cells can be identified using a series of criteria on the 1D track (Campbell et al., Nature Neuroscience, 2018; Campbell et al., Cell Reports, 2021). Thus, we are able to leverage the advantage of the 1D virtual reality track (behavioral consistency) and still consider the activity of grid cells.

3. On a related note, I was not certain why the authors chose to implement a 1D track for their virtual reality. I appreciate the simplicity of the behavior in this setting, but for the purposes of mapping grid and place cells, one does not necessarily need a behavioral task. The authors should comment on this.

We agree that a behavioral task is not necessarily needed to map grid cell activity. However, as noted above, we felt a 1D track combined with head-fixed virtual reality allowed us high behavioral consistency while also not sacrificing our ability to classify grid cells. Specifically, the reduced freedom of movement of a 1D track allows better comparison of spatial firing maps after ketamine as the mouse is forced to run along the same path, rather than spinning (see open field position traces in Supplementary Fig. 3). Mice do not walk spontaneously in a head-fixed environment, thus rewards were necessary to encourage movement. The behavioral task also revealed that mice stopped slowing down and stopped licking at reward locations acutely following ketamine administration, suggesting that they were less able to identify the reward location. We have text to the Results to comment on this.

4. This is a minor point, but I was not sure why the authors used both PCA and UMAP to cluster neuronal activity. It seems that UMAP is a bit better which is not terribly surprising. So, why do they also need PCA results?

We agree this is not necessary and have removed the PCA results.

REVIEWERS' COMMENTS

Reviewer #1 (Remarks to the Author):

Masuda and colleagues have very nicely revised their report on the effects of ketamine on the spatial coding characteristics of

hippocampal and entorhinal cortical neurons. They present profound effects on the spatial coding properties of these neurons. The work is elegant and the figures nicely demonstrate the observed phenomenon. The authors were very thorough in incorporating the suggestions/comments of both reviewers and from my perspective there is no need for additional changes.

Reviewer #2 (Remarks to the Author):

I appreciated the responses of the authors to the review. While I would have liked to see greater characterization of the dose-response curve. I am comfortable with the responses overall.

Reviewer #3 (Remarks to the Author):

My comments on the original manuscript were relatively minor and I believe that the authors dealt with them very well. Thank you. I found their discussion (in the rebuttal letter) of the limitation of the 1D track environment and the very interesting differences in grid cell activity in real and virtual tasks highly intriguing. I would encourage the authors to append the discussion section of the manuscript and include a couple of sentences that summarize the differences in MEC activity in real and virtual settings. There is a growing number of papers using virtual reality in rodents because of ease of recording neuronal activity but the differences between real and virtual settings have received considerably less attention.

Response to reviewer comments

We thank the reviewers for their constructive comments through the review process. We have addressed the final comment by R3 by including a few sentences at the end of the discussion regarding the challenges of examining spatial coding in 1D VR versus 2D freely moving tasks.